# NESTER: An Adaptive Neurosymbolic Method for Treatment Effect Estimation

## Abstract

Treatment effect estimation from observational data is a central problem in causal inference. Methods based on potential outcomes framework solve this problem by exploiting inductive biases and heuristics from causal inference. Each of these methods addresses a specific aspect of treatment effect estimation, such as controlling propensity score, enforcing randomization, etc., by designing neural network architectures and regularizers. In this paper, we propose an adaptive method called Neurosymbolic Treatment Effect Estimator (NESTER), a generalized method for treatment effect estimation. NESTER brings together the ideas used in existing methods based on multi-head neural networks for treatment effect estimation into one framework. To perform program synthesis, we design a Domain Specific Language (DSL) for treatment effect estimation based on inductive biases used in literature. We also theoretically study NESTER's capability for treatment effect estimation. Our comprehensive empirical results show that NESTER performs better than state-of-the-art methods on benchmark datasets without compromising run time requirements.

## 1 Introduction

Treatment effect (a.k.a. causal effect) estimation measures the effect of a treatment variable on an outcome variable (e.g., the effect of a medicine on recovery). Randomized Controlled Trials (RCTs), where individuals are randomly split into *treated* and *control (untreated)* groups, are considered the gold standard approach for treatment effect estimation (Chalmers et al., 1981; Pearl, 2009). However, RCTs are often: (i) unethical (e.g., in a study to find the effect of smoking on lung disease, a randomly chosen person cannot be forced to smoke), and/or (ii) impossible/infeasible (e.g., in finding the effect of blood pressure on the risk of an adverse cardiac event, it is impossible to intervene on the same patient with and without high blood pressure with all other parameters the same) (Sanson-Fisher et al., 2007; Carey & Stiles, 2016; Pearl et al., 2016). These limitations leave us with observational data to compute treatment effects.

Observational data, similar to RCTs, suffer from *the fundamental problem of causal inference* (Pearl, 2009), *viz.* for any individual, we cannot observe all potential outcomes at the same time (e.g., we cannot uniquely record the same person's medical condition/response at a given time to two different treatments individually, say, on consuming a medicinal drug and an alternate placebo). Observational data also suffers from *selection bias* (e.g., certain age groups are more likely to take certain kinds of medication compared to other age groups) (Collier & Mahoney, 1996). For these reasons, estimating unbiased treatment effects from observational data can be challenging (Hernan & Robins, 2019; Farajtabar et al., 2020). However, due to the many use cases in the real-world, estimating treatment effects from observational data has remained an important problem in causal inference (Rosenbaum & Rubin, 1983; 1985; Brady et al., 2008; Morgan & Winship, 2014), with recent efforts leveraging learning-based methods to this end (Curth & van der Schaar, 2021a; Zhang et al., 2021).

Simpson's paradox (Pearl et al., 2016) underpins the necessity of choosing the correct set of variables to *control/adjust* for estimating treatment effects from observational data. The Pearlian framework (Pearl, 2009) uses graphical criteria such as *back-door* criterion and *front-door* criterion depending on the available adjustment variables and identifiability conditions. However, the Pearlian framework requires knowledge of the underlying causal graph, which is not feasible for many real-world scenarios. On the other hand, under the *no latent confounding/ignorability* assumption, methods based on the classical *Neyman-Rubin potential outcomes framework* (Rubin, 1974) assume that a known set of observed features to control is available.

However, as discussed above, observational data suffers from issues such as selection bias, leading to biased estimates of treatment effects. Various methods have been proposed to address one or more of these issues in recent literature (Shalit et al., 2017; Shi et al., 2019; Farajtabar et al., 2020; Curth & van der Schaar, 2021a).

In this paper, we provide a pathway to integrate existing solutions based on the potential outcomes framework, especially the methods based on multi-head neural network (NN) architectures with regularizers, into a single framework and propose a generalized method for treatment effect estimation. As shown in Fig 1, our method generates programs, each of which can instantiate existing methods based on multi-head NN architectures with regularizers as special cases. Concretely, we propose an adaptive method called NEuroSymbolic Treatment Effect EstimatoR (NESTER) that automatically synthesizes different programs for estimating treatment effects given observational data. For example, the two branches (or heads) in the TARNet NN architecture (Shalit et al., 2017) in Fig 1 can be seen as implementing an if − then − else program primitive. That is, from Fig 1, in the synthesized program $\mathcal{P}_T$, if subset($\mathbf{v}, \{0\}$) evaluates to a positive scalar value (see § 4.1 for details), then clause gets executed otherwise else clause gets executed. Also, the IPM regularization in the counterfactual regression model (CFR) (Shalit et al., 2017) can be viewed as implementing the transform primitive and the propensity head $p(t|\phi)$ in Dragonnet Shi et al. (2019) can be seen as implementing subset primitive with the set of parents $\mathtt{pa_T}$ of the treatment variable $\mathtt{T}$ as a parameter (see § 4.1,4.2 for details).

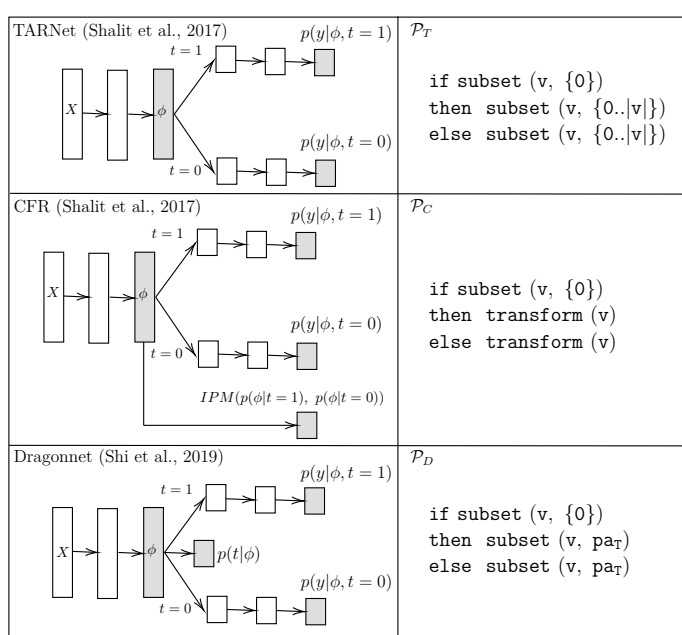

Figure 1: Programs $\mathcal{P}_T, \mathcal{P}_C, \mathcal{P}_D$ generated by NESTER using our Domain-Specific Language (DSL) (Tab 1) that are functionality similar to the popular multi-head NN models TARNet, CFR, and Dragonnet. $X$ is feature/covariate vector, $t$ is treatment variable, $y$ is target variable, $\phi$ is learned representation, $\mathbf{v} = [t; X]$ is the concatenation of $t, X$, $\mathtt{pa_T}$ is the parents of treatment variable.

As part of our proposed method NESTER, we develop a Domain-Specific Language (DSL) of program primitives (e.g., subset) containing learnable components, which is then used by a neurosymbolic program synthesis (NPS) technique to automatically synthesize programs. This is equivalent to putting together modules (program primitives in our DSL) to obtain a model architecture/workflow that can be used for the given observational data. In other words, NESTER learns to adaptively synthesize differentiable programs for a given set of input-output examples, wherein the sequence of learnable program modules provides an overall network architecture that is used to estimate treatment effects. Empirically, by limiting the depth of synthesized programs, NESTER performs state-of-the-art treatment effect estimation on benchmark datasets with almost no additional time overhead.

NPS methods generate programs using a language of program primitives that satisfy given observational data of input-output pairs so that the synthesized programs generalize well to unseen inputs (see Appendix § C,D for real-world examples) (Biermann, 1978; Gulwani, 2011; Parisotto et al., 2016; Valkov et al., 2018; Shah et al., 2020; Cui & Zhu, 2021). Usually, a Domain-Specific Language (DSL) (e.g., a specific *context-free grammar*) is used to synthesize programs for a given domain or task. Recently, various NN-based techniques have been proposed to perform NPS (Parisotto et al., 2016; Valkov et al., 2018; Gaunt et al., 2017; Bošnjak et al., 2017; Shi et al., 2022; Tang & Ellis, 2022; Cui & Zhu, 2021). We use an NPS paradigm where each program primitive (e.g., if − then − else, subset, add) is a differentiable module (Shah et al., 2020). Such *differentiable programs* simultaneously optimize program primitive parameters while learning the overall program structure. Many methods have been proposed to efficiently synthesize and learn such a program

using a DSL (Gulwani et al., 2012; Valkov et al., 2018; Shah et al., 2020). To synthesize programs, we use (i) *neural admissible relaxation* (NEAR) (Shah et al., 2020), which uses neural networks as relaxations of partial programs while searching the program space using informed search algorithms such as $A^*$ (Hart et al., 1968) and (ii) domain-specific program architecture differentiable synthesis (dPads) (Cui & Zhu, 2021) that learns the probability distribution of program architectures in a continuous relaxation of the search space of DSL grammar rules. Our key contributions in this work are summarized below:

- We develop an adaptive neurosymbolic method that can learn to estimate treatment effects given observational data. Such a method is not restricted by its architecture and is easy to implement and extend. To the best of our knowledge, this is the first neurosymbolic approach to estimate treatment effects.
- We propose a domain-specific language (DSL) for treatment effect estimation, whose program primitives are inspired by treatment effect estimation efforts in literature.
- We theoretically study the universal approximation ability of a synthesized neurosymbolic program and show how this provides a pathway for our method for treatment effect estimation. We also study how the proposed NESTER method can be viewed as a generalization of a class of treatment effect estimation methods based on multi-head NN architectures.
- We perform comprehensive empirical studies on multiple benchmark datasets where NESTER outperforms existing state-of-the-art models. We also observe that these results are obtained with almost no additional empirical time complexity beyond existing methods.

## 2 Related Work

**Matching and Covariate Adjustment Methods:** Early methods of treatment effect estimation are primarily based on matching techniques (Brady et al., 2008; Morgan & Winship, 2014; Stuart, 2010) where similar data points in treatment and control groups are compared using methods such as nearest neighbor matching and propensity score matching. In nearest neighbor matching (Stuart, 2010), for each sample in the treatment group, the nearest points from the control group w.r.t. Euclidean distance are identified, and the difference in observed outcomes between the treatment and corresponding control data points is the estimate of treatment effect. In propensity score matching (Rosenbaum & Rubin, 1983), a model is trained to predict the treatment value using data from both treatment and control groups. Using this model, points from treatment and control groups that are close w.r.t. the model's output are compared, and the difference in observed outcomes of these points is the estimate of treatment effect. However, such matching techniques are known to not scale to high-dimensional or large-scale data (Abadie & Imbens, 2006).

Another family of methods estimates treatment effects using the idea of backdoor adjustment (Pearl, 2009; Rubin, 2005). Assuming the availability of a sufficient adjustment set, these models rely on fitting conditional probabilities given the treatment variable and a sufficient adjustment set of covariates. Such models are however known to suffer from high variance in the estimated treatment effects (Shalit et al., 2017). Covariate balancing is another technique to control for the confounding bias in estimating treatment effects. Weighting techniques perform covariate balancing by assigning weights to each instance based on various techniques (e.g., weighting each instance using propensity score in the inverse probability weighting technique) (Rosenbaum & Rubin, 1983; Assaad et al., 2021; CRUMP et al., 2009; Li & Greene, 2013; Diamond & Sekhon, 2013; Li & Fu, 2017). As noted in Assaad et al. (2021), such methods face challenges with large weights and high-dimensional inputs. Besides, leveraging the success of learning-based methods has yielded significantly better performance in recent years.

**Representation Learning-based Methods:** Recent methods to estimate treatment effects have largely been based on multi-head NN architectures (NN architectures which branch out into different heads for different treatments) equipped with regularizers (Curth & van der Schaar, 2021a;b; Shi et al., 2019; Schwab et al., 2020; Chu et al., 2020; Shalit et al., 2017). Considering multiple treatment values and continuous dosage for each treatment, Schwab et al. (2020) devised an NN architecture with multiple heads for multiple treatments, and multiple sub-heads from each of the treatment-specific heads to model (discretized) dosage values. CFR (Shalit et al., 2017) proposed a two-headed NN architecture with a regularizer that forced the latent representations of treatment and control groups to be close to each other to adjust confounding features. Extending CFR, (Farajtabar et al., 2020) proposed an additional regularizer to adjust for confounding by forcing both treatment-specific heads to have same baseline outcomes. In Dragonnet (Shi et al., 2019),

along with two heads for predicting treatment-specific (potential) outcomes, an additional head to predict treatment value was also used; this allowed pre-treatment covariates to be used in predicting potential outcomes. Assuming that potential outcomes are strongly related, (Curth & van der Schaar, 2021a;b) proposed techniques that improve existing models using the structural similarities between potential outcomes. These methods, however, have a fixed architecture design, and each addresses a specific problem in estimating treatment effects. Our approach is also NN-based but uses a neurosymbolic approach to automatically synthesize an architecture (or a flow of program primitives), thereby allowing it to generate different programs for different observational data. Generative Adversarial Networks (GANs) (Goodfellow et al., 2014) have also been used to learn the interventional distribution (Yoon et al., 2018; Bica et al., 2020) from observed data in both categorical and continuous treatment variable settings to estimate treatment effects. By disentangling confounding variables from instrumental variables, Zhang et al. (2021) proposed a variational inference method that uses only confounding variables. However, generative modeling requires a large amount of data to be useful, which is often not practical in treatment effect estimation tasks.Yao et al. (2018) proposed a method to learn representations by leveraging local similarities and thereby estimate treatment effects. Ensemble models such as causal forests (Wager & Athey, 2018), and Bayesian additive regression trees (Chipman et al., 2010) have also been considered for effect estimation. Our work is very different from these efforts – we choose to integrate the heuristics and corresponding NN architectures under a single framework, and seek to provide a flexible yet powerful framework for treatment effect estimation using NPS.

**Relevance of Causal Discovery Methods:** In Pearlian approaches to treatment effect estimation (which assume knowledge of causal graph), performing *causal discovery* before treatment effect estimation has also been studied in literature (Hoyer et al., 2008; Mooij et al., 2016; Maathuis et al., 2010; Gupta et al., 2022). However, NN-based learning approaches have primarily focused on the potential outcomes approach for this objective. Since our work is situated in the latter context, under the *ignorability* assumption (see § 3), we assume that the underlying causal graph takes a form in which the treatment is independent of the potential outcomes given a set (possibly empty) of pre-treatment covariates (Shalit et al., 2017). Thus, we avoid performing causal discovery before estimating treatment effects.

**Neurosymbolic Program Synthesis (NPS):** Program synthesis, *viz.* automatically learning a program that satisfies given observational data of input-output pairs (Biermann, 1978; Gulwani, 2011; Parisotto et al., 2016; Valkov et al., 2018; Shah et al., 2020; Shi et al., 2022; Tang & Ellis, 2022; Cui & Zhu, 2021), has been shown to be helpful in diverse tasks such as low-level bit manipulation code (Solar-Lezama et al., 2005), data structure manipulations (Solar Lezama, 2008), and regular expression-based string generation (Gulwani, 2011). For each task, a specific DSL is used to synthesize programs. Even with a small DSL, the number of programs that can be synthesized is very large. Several techniques such as greedy enumeration, Monte Carlo sampling, Monte Carlo tree search (Kocsis & Szepesvári, 2006), evolutionary algorithms (Valkov et al., 2018), and recently, node pruning with neural admissible relaxation (NEAR) (Shah et al., 2020) have been proposed to search for optimal programs from a vast search space efficiently. Improving NEAR, dPads Cui & Zhu (2021) propose differentiable program architecture synthesis that avoids the problem of combinatorial search required to find optimal programs. We use both NEAR and dPads to implement our method. To the best of our knowledge, this is the first use of NPS for treatment effect estimation.

**Neurosymbolic Program Synthesis (NPS) vs Neural Architecture Search (NAS):** NAS, similar to NPS, is a technique to automatically design the best NN architecture to solve particular problems (Zoph & Le, 2017; Liu et al., 2018; Pham et al., 2018). The significant difference between NPS and NAS is that, in NPS, symbolic and domain knowledge can be introduced in terms of the program primitives of a DSL. However, the goal in NAS is to design the best-performing architecture using a combination of standard neural network components, such as convolution blocks. In NPS, DSL changes for different applications, whereas in NAS, the underlying neural network blocks are fixed for all the problems. In NPS, we can combine symbolic reasoning and representation learning algorithms, making it a good choice for treatment effect estimation.

## 3 Background and Problem Formulation

**Treatment Effect Estimation:** Let $\mathcal{D} = \{(\mathbf{x}_i, t_i, y_i)\}_{i=1}^{n}$ be an observational dataset of $n$ triplets. $\mathbf{x}_i \in \mathbb{R}^d$ denotes the $d-$dimensional covariate vector, $t_i \in \mathbb{R}$ denotes the treatment variable ($t_i \not\subseteq \mathbf{x}_i$), and $y_i \in \mathbb{R}$ denotes the observed potential outcome. Each $(\mathbf{x}_i, t_i, y_i)$ is randomly sampled from $p(\mathbf{X}, T, Y)$, where $\mathbf{X}$,

$Y$ and $T$ are the corresponding random variables. In a binary treatment setting ($t \in \{0,1\}$), for the $i^{th}$ observation, let $Y_i^0$ denote the true potential outcome under treatment $t_i = 0$ and $Y_i^1$ denote the true potential outcome under treatment $t_i = 1$. Because of *the fundamental problem of causal inference*, we observe only one of $Y_i^0, Y_i^1$ for a given $[t_i; \mathbf{x}_i]$. Hence, observed $y_i$ can be expressed in terms of $Y_i^0, Y_i^1$ as $y_i = t_i Y_i^1 + (1 - t_i) Y_i^0$. One of the goals in treatment effect estimation from observational data is to learn an estimator $f(\mathbf{x}, t)$ such that the difference between estimated potential outcomes under $t = 1$ and $t = 0$, $f(\mathbf{x}_i, 1) - f(\mathbf{x}_i, 0)$, is close to the difference in true potential outcomes: $Y_i^1 - Y_i^0$; $\forall i$. This difference for a specific instance/individual $i$ is called the *Individual Treatment Effect (ITE)*.

**Definition 3.1.** *The Individual Treatment Effect (ITE) of $T$ on $Y$ for an instance $\mathbf{x} \sim \mathbf{X}$ is defined as*

$$ITE_T^Y(\mathbf{x}) := \mathbb{E}[Y^1 - Y^0|\mathbf{x}] \tag{1}$$

**Definition 3.2.** *With respect to the true $ITE_T^Y(\mathbf{x})$, the expected Precision in Estimation of Heterogeneous Effect ($\epsilon_{PEHE}$) using $f(\mathbf{x}, t)$ is defined as*

$$\epsilon_{PEHE}(f) := \mathop{\mathbb{E}}_{\mathbf{x} \sim \mathbf{X}}\left[\left((f(\mathbf{x}, 1) - f(\mathbf{x}, 0)) - ITE_T^Y(\mathbf{x})\right)^2\right] \tag{2}$$

Extending *ITE* to an entire population, our goal is to estimate the *Average Treatment Effect (ATE)* (Pearl, 2009) of the treatment variable $T$ on the outcome variable $Y$.

**Definition 3.3.** *The Average Treatment Effect (ATE) of $T$ on $Y$ is defined as*

$$ATE_T^Y := \mathbb{E}[Y|do(T = 1)] - \mathbb{E}[Y|do(T = 0)] \tag{3}$$

**Definition 3.4.** *The error in estimation of Average Treatment Effect ($\epsilon_{ATE}$) using $f(\mathbf{x}, t)$ is defined as*

$$\epsilon_{ATE}(f) := |\mathop{\mathbb{E}}_{\mathbf{x} \sim \mathbf{X}}[f(\mathbf{x}, 1) - f(\mathbf{x}, 0)] - ATE_T^Y| \tag{4}$$

$do(.)$ in Defn 3.3 denotes an intervention to the treatment variable (Pearl, 2009). $\mathbb{E}[Y|do(T = t)]$ refers to the expected value of $Y$ when every instance in the population is given the treatment $t$ (if $t$ is not binary, treatment effects are calculated w.r.t. a baseline treatment value $t^*$ i.e., $1, 0$ in Defn 3.3 are replaced with $t, t^*$ respectively). Assuming $\mathbf{X}$ satisfies the backdoor criterion relative to the treatment effect of $T$ on $Y$, we can write $\mathbb{E}[Y|do(T = t)] = \mathbb{E}_{\mathbf{x} \sim \mathbf{X}}[\mathbb{E}[Y|T = t, \mathbf{X} = \mathbf{x}]]$ (Pearl, 2009). Using this, a simple technique to estimate $\mathbb{E}[Y|T = t, \mathbf{X} = \mathbf{x}]$ (and thus $\mathbb{E}[Y|do(T = t)]$) is to fit a regression model for $Y$ given $T$, and $\mathbf{X}$. Estimation of $\mathbb{E}[Y|T = t, \mathbf{X} = \mathbf{x}]$ is the primary task of many treatment effect estimation methods. In the same vein, we also aim to synthesize programs that compute the quantity $\mathbb{E}[Y|T = t, \mathbf{X} = \mathbf{x}]$ given the observational data $\mathcal{D}$. Once $\mathbb{E}[Y|T = t, \mathbf{X} = \mathbf{x}]$ is estimated, the $ATE_T^Y$ can be estimated as $ATE_T^Y = \mathbb{E}_{\mathbf{x} \sim \mathbf{X}}[\mathbb{E}[Y|T = 0, \mathbf{X} = \mathbf{x}]] - \mathbb{E}_{\mathbf{x} \sim \mathbf{X}}[\mathbb{E}[Y|T = 0, \mathbf{X} = \mathbf{x}]]$. Following (Shalit et al., 2017; Lechner, 2001; Imbens, 2000; Schwab et al., 2020; Zhang et al., 2021), we make the following assumptions which are sufficient to guarantee the *identifiability* (Pearl, 2009) of treatment effects from observational data.

**Assumptions 3.1.** *(Ignorability, Positivity, Stable Unit Treatment Value Assumption (SUTVA)) Ignorability says that for a given set of pre-treatment covariates, treatment is randomly assigned, i.e., conditioned on a set of pre-treatment covariates $\mathbf{X}$, $T$ is independent of $Y^0, Y^1$ i.e., $((Y^0, Y^1) \perp\!\!\!\perp T|\mathbf{X})$. Positivity entails that treatment assignment for each individual is not deterministic, and it must be possible to assign all treatments to each individual, i.e., $0 < p(t|\mathbf{x}) < 1 \ \forall t, \mathbf{x}$. SUTVA states that the observed outcome of any individual under treatment must be independent of the treatment assignment to other individuals. (Ignorability is also referred to as no-latent-confounding assumption.)*

**Neurosymbolic Program Synthesis (NPS):** Following Shah et al. (2020), let $(\mathcal{P}, \theta)$ be a neurosymbolic *program* where $\mathcal{P}$ denotes the program *structure* and $\theta$ denotes the program *parameters*. $(\mathcal{P}, \theta)$ is differentiable in $\theta$. $\mathcal{P}$ is synthesized using a Context-Free Grammar (CFG) (Hopcroft et al., 2001) (which is a DSL in this work). A CFG consists of a set of *production rules* (rules for short) of the form $\rho \rightarrow \sigma_1, \dots, \sigma_n$ where $\rho$ is a *non-terminal* and $\sigma_1, \dots, \sigma_n$ are either *non-terminals* or *terminals*. A nonterminal denotes a missing subexpression in a program structure and a terminal is a symbol that appears in a final program structure. Program synthesis starts with an initial non-terminal (see Fig 2), then iteratively applies the production rules to produce a series of *partial structures*, viz. structures made from one or more non-terminals and zero or more terminals. These partial structures form internal nodes of a *program tree*, and the production rules

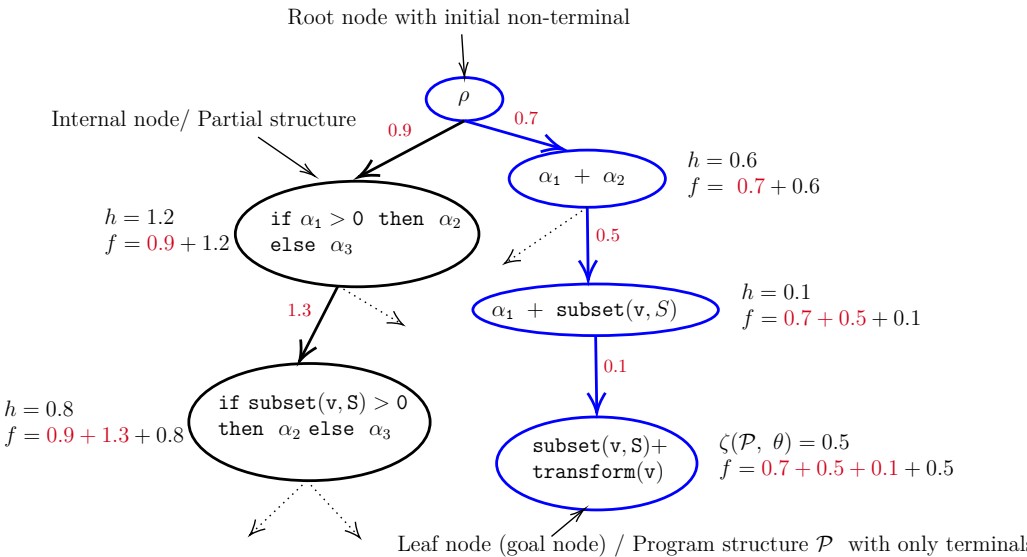

Figure 2: Example program tree generated using DSL in Tab 1. Structural costs are shown in red color (e.g., $s(r) = 0.7$ for the rule $r : \rho \to \alpha_1 + \alpha_2$). $h$ is the heuristic value, and $f$ is the sum of structural cost and heuristic value. The path from the root node to a goal node returned by $A^*$ algorithm is shown in blue color. For this example, $s(r)$ is not constant.

form the (directed) edges connecting these nodes (e.g., a production rule $r$ is considered as an edge from node $u$ to node $v$ when $v$ is obtained from $u$ by applying $r$). The process continues until no non-terminals are left, i.e., we have synthesized a program. The resultant program tree's leaf nodes (a.k.a. goal nodes) contain structures consisting of only terminals (see Fig 2 for an example).

Let $s(r)$ be the cost incurred in using the production rule $r$ for generating a program structure (leaf node) or partial structure (internal node) from a given partial structure (internal node). The structural cost of any node $u$ is the sum of the structural costs of rules used to get $u$ from the root node. Similarly, the structural cost $s(\mathcal{P})$ of the program $\mathcal{P}$ is defined as $s(\mathcal{P}) = \sum_{r \in R(\mathcal{P})} s(r)$, where $R(\mathcal{P})$ is the multiset of rules used to create the structure $\mathcal{P}$. In this paper, we set $s(r)$ to a constant real number for all production rules (e.g., $s(r) = 1 \ \forall r \in R(\mathcal{P})$). The program learning problem is usually formulated as a node search problem, i.e., starting with an empty tree, the tree is expanded by creating new partial structures (internal nodes) and structures (leaf nodes).

This paper uses $A^*$ informed search algorithm Hart et al. (1968) to generate a program tree. While generating a program tree, a node $u$ with minimum $f(u)$ value is expanded next, where $f(u) = s(\mathcal{P}(u)) + h(u)$ is the sum of the structural cost $s(\mathcal{P}(u))$ of the partial structure $\mathcal{P}(u)$ in $u$ plus the heuristic value $h(u)$ at the node $u$ that underestimates the cost to reach goal node from $u$ (see Fig 2 and § 4.3). While searching for an optimal program, the program parameters (and program structures) are updated simultaneously along with the synthesis of the programs. Since the goal of this paper is to synthesize a program $(\mathcal{P}, \theta)$ that estimates the quantity $\mathbb{E}[Y|T = t, \mathbf{X} = \mathbf{x}]$, which can be modeled as a regression problem, the squared error is a good choice for assessing the performance of the program $(\mathcal{P}, \theta)$ in estimating potential outcomes. Hence, for a synthesized program $(\mathcal{P}, \theta)$, we define $\zeta(\mathcal{P}, \theta) = \mathbb{E}_{(\mathbf{x}, t, y) \sim \mathcal{D}}[((\mathcal{P}, \theta)(\mathbf{x}, t) - y)^2]$ as the loss incurred by $(\mathcal{P}, \theta)$ in estimating potential outcomes. The overall goal of NPS is then to find a structurally simple program with low prediction error, i.e., to solve the following optimization problem.

$$(\mathcal{P}^*, \theta^*) = \arg\min_{(\mathcal{P}, \theta)} (s(\mathcal{P}) + \zeta(\mathcal{P}, \theta)) \tag{5}$$

## 4 NESTER: Methodology

The key idea of our methodology is to design a Domain-Specific Language (DSL) for treatment effect estimation and subsequently leverage well-known search algorithms such as $A^*$ to synthesize programs or model architectures for given observational data. We begin by discussing the proposed DSL and its connections to existing literature, followed by our overall algorithm that uses this DSL to synthesize programs.

### 4.1 DSL for Treatment Effect Estimation

We pose the problem of treatment effect estimation as the problem of mapping a set of observational input data points to the corresponding observed outcomes. Formally, given $\mathcal{D} = \{(\mathbf{x}_i, t_i, y_i)\}_{i=1}^n$, the set $\{(t_i, \mathbf{x}_i)\}_{i=1}^n$ contains inputs and the set $\{y_i\}_{i=1}^n$ contains outputs. For simplicity, let $\mathbf{v}_i = [t_i; \mathbf{x}_i]$ (concatenation of $t_i$ and $\mathbf{x}_i$) denote the $i^{th}$ input. A synthesized program learns to estimate the potential outcomes for unseen inputs by learning a mapping between given input-output examples. To this end, we propose a set of program primitives (basic building blocks of a synthesized program), which are differentiable and encode specific inductive biases in an NN model architecture. These primitives comprise our proposed DSL, shown in Tab 1. Each of these listed primitives outputs a real scalar number, which can be the final output (terminal) or fed as input into another primitive. We briefly describe each of them below. We also later state Propn 5.1 in § 5 that guarantees the existence of a DSL for the treatment effect estimation task.

Table 1: A DSL for the treatment effect estimation in Backus-Naur form (Winskel, 1993) and its semantics. $\rho$ is the initial non-terminal. $\mathbf{v} = [t; \mathbf{x}]$ is input from $\mathcal{D}$. MLP stands for multi-layer perceptron. All primitives output real numbers as output.

$$\rho \rightarrow \texttt{if } \alpha_1 > 0 \texttt{ then } \alpha_2 \texttt{ else } \alpha_3 \mid \texttt{subset}(\mathbf{v}, \texttt{S}) \mid \texttt{transform}(\mathbf{v}) \mid \odot(\alpha_1, \alpha_2)$$
$$\alpha_1/\alpha_2/\alpha_3 \rightarrow \texttt{if } \alpha_1 > 0 \texttt{ then } \alpha_2 \texttt{ else } \alpha_3 \mid \texttt{subset}(\mathbf{v}, \texttt{S}) \mid \texttt{transform}(\mathbf{v}) \mid \odot(\alpha_1, \alpha_2)$$

| Program Primitive | Description |
|---|---|
| 1. if $\alpha_1 > 0$ then $\alpha_2$ else $\alpha_3$ | Usual $\texttt{if} - \texttt{then} - \texttt{else}$ condition. To avoid discontinuities and to enable backpropagation, we implement a smooth approximation of $\texttt{if} - \texttt{then} - \texttt{else}$. |
| 2. $\texttt{subset}(\mathbf{v}, \texttt{S})$ | Select/retain a set of features of $\mathbf{v}$ indexed by the set $S$. Features at other indices are set to 0. Feed the resultant vector into an MLP to get a real number as output. |
| 3. $\texttt{transform}(\mathbf{v})$ | Transforms the input vector $\mathbf{v}$ into $\phi(\mathbf{v})$ using a pre-trained model $\phi$ as explained in § 4.1. Feed $\phi(\mathbf{v})$ into an MLP to get a real number as output. |
| 4. $\odot(\alpha_1, \alpha_2)$ | Arithmetic function of $\alpha_1, \alpha_2$ where $\odot \in \{+, -, \times, /\}$ (e.g., $\alpha_1 + \alpha_2, \alpha_1 \times \alpha_2$). |

The primitive "$\texttt{if } \alpha_1 > 0 \texttt{ then } \alpha_2 \texttt{ else } \alpha_3$" works similar to the equivalent programming construct. To avoid discontinuities and enable backpropagation, following Shah et al. (2020), we implement a smooth approximation of $\texttt{if} - \texttt{then} - \texttt{else}$, i.e., $\texttt{if } \alpha_1 > 0 \texttt{ then } \alpha_2 \texttt{ else } \alpha_3$ can be written as $\texttt{sig}(\beta \cdot \alpha_1) \cdot \alpha_2 + (1 - \texttt{sig}(\beta \cdot \alpha_1)) \cdot \alpha_3$, where $\texttt{sig}(\cdot)$ is the *sigmoid* function and $\beta$ is a temperature parameter. As $\beta \rightarrow 0$, the approximation approaches the usual $\texttt{if} - \texttt{then} - \texttt{else}$. Since we implement a smooth approximation of $\texttt{if} - \texttt{then} - \texttt{else}$, it is not required for $\alpha_1$ to evaluate to a boolean value.

The primitive "$\texttt{subset}(\mathbf{v}, \texttt{S})$" selects a set of features of $\mathbf{v}$ indexed by the set $\texttt{S}$ of indices. Other features of $\mathbf{v}$ that are not indexed by the set $S$ are set to 0. The resultant vector is then fed into a multi-layer perceptron (MLP) (whose parameters are learned during the end-to-end backpropagation of the full program) to get a real number as output.

The "$\texttt{transform}(\mathbf{v})$" primitive transforms a given input vector $\mathbf{v}$ into $\phi(\mathbf{v})$. $\phi$ is an NN whose parameters are optimized to produce similar outputs for inputs with different treatment values to act as a regularizer based on the Integral Probability Metric (IPM), similar to Shalit et al. (2017). In particular, given two inputs $\mathbf{v}_0 \sim p(\mathbf{v}|t=0)$ and $\mathbf{v}_1 \sim p(\mathbf{v}|t=1)$, we would want $\phi(\mathbf{v}_0) \approx \phi(\mathbf{v}_1)$. We pre-train $\phi$ such that the Maximum Mean Discrepancy is minimized between $p(\phi(\mathbf{v})|t=1)$ and $p(\phi(\mathbf{v})|t=0)$. Since $\phi$ is pre-trained, all instances of $\texttt{transform}$ in a synthesized program share the same representation of the pre-trained model $\phi$. The transformed vector $\phi(\mathbf{v})$ is subsequently fed into a learnable MLP to produce a real number as output. Even though $\phi$ seems like the backbone in Shalit et al. (2017), unlike its fixed architecture, the proposed program synthesizer can choose when to use this primitive. The synthesizer can also use $\texttt{transform}$ multiple times in a program too (see Tab 5 for examples).

The last primitive, "$\odot(\alpha_1, \alpha_2)$" is included for giving additional flexibility to the program synthesizer to allow simple arithmetic operations. $\odot(\alpha_1, \alpha_2)$ takes two real numbers as inputs and returns a real number as output after performing an arithmetic operation $\odot$.

## 4.2 Connection to Existing Methods

As discussed earlier, existing learning-based treatment effect estimation methods introduce inductive biases into machine learning models through regularizers or through changes in NN architectures. One could view the primitives of our DSL as learnable modules inspired by existing learning-based methods such as TARNet, CFR (Shalit et al., 2017), Dragonnet (Shi et al., 2019), SNet (Curth & van der Schaar, 2021a), etc. Tab 2 presents a summary of these relationships, which we also discuss below.

$(\texttt{if} - \texttt{then} - \texttt{else}, \texttt{subset})$ **Primitives for Multi-head NNs:** In treatment effect estimation, our goal is to estimate the quantity $\mathbb{E}[Y|T = t, \mathbf{X} = \mathbf{x}]$. If a single model is used to estimate both $\mathbb{E}[Y|T = 0, \mathbf{X} = \mathbf{x}]$ and $\mathbb{E}[Y|T = 1, \mathbf{X} = \mathbf{x}]$, it is often the case that $\mathbf{X}$ is high-dimensional and the treatment $T$ is a relatively much smaller set of variables (often, just one variable) when compared to $\mathbf{X}$.

Hence, $T$ may not have an impact on the model when making predictions, resulting in the estimated treatment effect being biased towards zero (Künzel et al., 2019). Using two different models to estimate $\mathbb{E}[Y|T = 0, \mathbf{X} = \mathbf{x}]$ and $\mathbb{E}[Y|T = 1, \mathbf{X} = \mathbf{x}]$ suffers from high variance in estimating treatment effect due to limited data in treatment-specific sub-groups as well as from selection bias. Shalit et al. (2017); Shi et al. (2019); Schwab et al. (2020); Farajtabar et al. (2020) hence leverage

Table 2: Connection between inductive biases in existing literature and the program primitives in the proposed DSL.

| Regularizer/Architectural Changes | Alternative Primitives |
|---|---|
| Two-head/Multi-head network (Farajtabar et al., 2020),(Shi et al., 2019) (Shalit et al., 2017),(Schwab et al., 2020) | $\texttt{if} - \texttt{then} - \texttt{else}$ $\texttt{subset}$ |
| Pre-treatment selection, Propensity Score Matching(Shi et al., 2019) | $\texttt{subset}$ |
| IPM regularization (Shalit et al., 2017),(Farajtabar et al., 2020) | $\texttt{transform}$ |

modified NN architectures in which two separate heads are spawned from a latent representation layer (See Fig 1) to predict treatment-specific outcomes. To implement such a two-head NN architecture, an NPS can leverage the $\texttt{if} - \texttt{then} - \texttt{else}$ and $\texttt{subset}$ primitives. As in Tab 1, replacing $\alpha_1$ with an appropriate $\texttt{subset}(\mathbf{v}, 0)$ would check for the treatment variable value, and accordingly return $\alpha_2$ or $\alpha_3$, which in turn are each sub-structures that act as two heads of the overall architecture. A NN with multiple heads would be implemented using nested $\texttt{if} - \texttt{then} - \texttt{else}$ primitives. We reiterate that we do not hard-code/pre-define the network architecture; the NPS learns to generate programs that compose primitives suitably to minimize overall loss during training.

$\texttt{subset}$ **Primitive for Covariate Selection:** To achieve *ignorability*, pre-treatment covariates are typically controlled while estimating treatment effect (e.g.,Dragonnet (Shi et al., 2019) controls pre-treatment covariates via controlling propensity score). However, controlling all input covariates may not be required. To identify a correct set of pre-treatment covariates to control, we can use the $\texttt{subset}(\mathbf{v}, \texttt{S})$ primitive. If we are unsure on the specific covariates, multiple instances of $\texttt{subset}(\mathbf{v}, \texttt{S})$ with different $\texttt{S}$ can be used, allowing the NPS to select the appropriate subset for given data.

$\texttt{transform}$ **Primitive for IPM Regularization:** To improve the results from two-head NN models, CFR (Shalit et al., 2017) used *IPM regularization* (using Maximum Mean Discrepancy (Gretton et al., 2012) or Wasserstein distance (Cuturi & Doucet, 2014)) on a latent layer representation. As in § 4.1, the $\texttt{transform}$ primitive is intended to achieve a similar purpose in our DSL. We now present the algorithm to synthesize neurosymbolic programs for the estimation of treatment effects.

## 4.3 Overall Algorithm

We refer to § 3 for the background on NPS, which we build on here. We use the $A^*$ informed search algorithm (Hart et al., 1968) to implement NESTER. At any internal node $u$, informed search algorithms usually rely on a *heuristic value $h(u)$* that underestimates the cost to reach the goal node from $u$. $h(u)$ decides which node to explore/expand next. During program tree generation, non-terminals in an internal node $u$ are substituted by a type-correct NN or MLP (e.g., if the primitive $\alpha_1 + \alpha_2$ returns a vector as output, the NNs substituted for $\alpha_1, \alpha_2$ must also return vectors as outputs). The training loss of the resultant program $(\mathcal{P}(u), \theta(u))$ on $\mathcal{D}$ then acts as the heuristic value $h(u)$ at node $u$ (Shah et al., 2020). We run the $A^*$ algorithm using this heuristic function to find programs that estimate treatment effects. We outline our overall algorithm in Algorithm 1. Algorithm 1 returns the program that satisfies the objective 5. Similar to

traditional NN training, the parameters of the best program (line 10 of Algorithm 1) are chosen based on the cross-validation score. Using a small DSL and keeping the overall program to only a limited depth allows us to build models that are efficiently learned and effective in practice with almost no additional time overhead compared to the state-of-the-art methods.

---

**Algorithm 1** NESTER using $A^*$

---

**Require:** Root node $u_0$ with initial non-terminal, DSL $\mathcal{L}$.
1: **Initialize:** $Q = \{u_0\}$, $f(u_0) = \infty$, $u_{best} = u_0$, $f_{best} = \infty$
2: **while** $Q \neq \emptyset$ **do**
3:     $v \leftarrow \arg\min_{u \in Q} f(u), \quad Q \leftarrow Q \setminus \{v\}$            ▷ $Q$ contains unexplored nodes in search
4:     **if** $v$ is leaf node and $f(v) < f_{best}$ **then**           ▷ $v$ contains only terminals
5:         $f_{best} \leftarrow f(v), \quad u_{best} \leftarrow v$
6:     **else**
7:         **for** child $u$ of $v$ **do**         ▷ Create new partial structures from $v$ using DSL $\mathcal{L}$
8:             $h(u) \leftarrow \min_{\theta(u)} \zeta(\mathcal{P}(u), \theta(u))$     ▷ $\mathcal{P}(u)$ contains MLPs in place of non-terminals in $u$
9:             $f(u) \leftarrow s(\mathcal{P}(u)) + h(u)$               ▷ $s$ is defined in § 3
10:             $Q \leftarrow Q \cup \{u\}$
11:         **end for**
12:     **end if**
13: **end while**
14: **return** $u_{best}$

---

## 5 NESTER: Analysis

We analyze NESTER from two perspectives: (i) the capability of a program synthesized using NPS methods to achieve treatment effect estimation, and (ii) the capabilities of our proposed DSL in relation to well-known learning-based treatment effect estimation methods. For the former, we hypothesize that if the relationship between treatment and effect is a continuous function, NPS is a viable candidate for estimating treatment effects. To this end, we first define the notion of an $\epsilon$-admissible heuristic in Defn 5.1, show how a synthesized program's training loss can serve as such an $\epsilon$-admissible heuristic in Lemma 5.1, and then state our result in Propn 5.1. All proofs are in Appendix § A.

**Definition 5.1.** *($\epsilon$-Admissible Heuristic (Harris, 1974; Pearl, 1984)) In an informed search algorithm, a heuristic function $h(u)$ that estimates the cost to reach the goal node $g$ from a node $u$ is said to be admissible if $h(u) \leq h^*(u), \forall u$ where $h^*(u)$ is the true cost to reach $g$ from $u$. Given an $\epsilon > 0$, $h(u)$ is said to be $\epsilon-$admissible if $h(u) \leq h^*(u) + \epsilon, \forall u$.*

**Lemma 5.1.** *(Neural Admissible Relaxations (Shah et al., 2020)) In an informed search algorithm $\mathcal{A}$, given an internal node $u_i$ and a leaf node $u_l$, let the cost of the leaf edge $(u_i, u_l)$ be $s(r) + \zeta(\mathcal{P}, \theta^*)$, where $\theta^* = \arg\min_\theta \zeta(\mathcal{P}, \theta)$ and $s(r)$ is the structural cost in using rule $r$ to create $u_l$ from $u_i$. If a neural network model $\mathcal{N}$ with adequate capacity in terms of depth and width of $\mathcal{N}$ is used to substitute each non-terminal of $u_i$, the training loss of the program obtained is an $\epsilon-$admissible heuristic for $u_i$.*

**Proposition 5.1.** *(Universal Approximation Result for NPS) If the interventional effect of the treatment variable $T$ on the target variable $Y$ is a continuous function $g(T)$ i.e., $\mathbb{E}[Y|do(T)] = g(T)$, using any DSL $\mathcal{L}$ for synthesizing a single-hidden-layer neural network, NESTER synthesizes a program $(\mathcal{P}, \theta)$ that $\epsilon-$approximates $g$ for a given $\epsilon > 0$.*

Our proof follows from the universal approximation theorem for NN models (Hornik et al., 1989), a DSL for a single-hidden-layer NN and Lemma 5.1. The above result shows that if the relationship between treatment and effect is a continuous function, using NESTER is a viable candidate for estimating treatment effects. We next discuss the capabilities of the proposed DSL w.r.t. existing methods.

**Proposition 5.2.** *(Error Bounds of NESTER) The program $(\mathcal{P}_C, \theta_C)$ generated by NESTER using the proposed DSL, whose architecture is the same as CFR (Shalit et al., 2017), has the same error bounds in estimating treatment effects as that of CFR.*

The above theoretical results show that the models for treatment effect estimation generated by NESTER can be shown to have performance bounds for the task similar to existing methods.

Table 4: Results on IHDP, Twins, and Jobs datasets. Lower is better. The best numbers are in bold. Simple machine learning models, ensemble models, and neural network-based models are separated using horizontal lines. Further analysis on k-NN results and dataset details are in Appendix § B

| Datasets (Metric) → | IHDP ($\epsilon_{ATE}(\downarrow)$) | | Twins ($\epsilon_{ATE}(\downarrow)$) | | Jobs ($\epsilon_{ATT}(\downarrow)$) | |
|---|---|---|---|---|---|---|
| Methods ↓ | In-Sample | Out-of-Sample | In-Sample | Out-of-Sample | In-Sample | Out-of-Sample |
| OLS-1 | .73±.04 | .94±.05 | .0038±.0025 | .0069±.0056 | **.01±.00** | .08±.04 |
| OLS-2 | .14±.01 | .31±.02 | .0039±.0025 | .0070±.0059 | **.01±.01** | .08±.03 |
| k-NN | .14±.01 | .90±.05 | .0028±.0021 | .0051±.0039 | .21±.01 | .13±.05 |
| BLR (Johansson et al., 2016) | .72±.04 | .93±.05 | .0057±.0036 | .0334±.0092 | **.01±.01** | .08±.03 |
| BART (Chipman et al., 2010) | .23±.01 | .34±.02 | .1206±.0236 | .1265±.0234 | .02±.00 | .08±.03 |
| Random Forest (Breiman, 2001) | .73±.05 | .96±.06 | .0049±.0034 | .0080±.0051 | .03±.01 | .09±.04 |
| Causal Forest (Wager & Athey, 2018) | .18±.01 | .40±.03 | .0286±.0035 | .0335±.0083 | .03±.01 | .07±.03 |
| BNN (Johansson et al., 2016) | .37±.03 | .42±.03 | .0056±.0032 | .0203±.0071 | .04±.01 | .09±.04 |
| TARNet (Shalit et al., 2017) | .26±.01 | .28±.01 | .0108±.0017 | .0151±.0018 | .05±.02 | .11±.04 |
| MHNET (Farajtabar et al., 2020) | .14±.13 | .37±.43 | .0108±.0008 | .0101±.0002 | .04±.01 | .06±.02 |
| GANITE (Yoon et al., 2018) | .43±.05 | .49±.05 | .0058±.0017 | .0089±.0075 | **.01±.01** | .06±.03 |
| CFR$_{WASS}$ (Shalit et al., 2017) | .25±.01 | .27±.01 | .0112±.0016 | .0284±.0032 | .04±.01 | .09±.03 |
| Dragonnet (Shi et al., 2019) | .16±.16 | .29±.31 | .0057±.0003 | .0150±.0003 | .04±.00 | .04±.00 |
| CMGP (Alaa & van der Schaar, 2017) | .11±.10 | .13±.12 | .0124±.0051 | .0143±.0116 | .06±.06 | .09±.07 |
| TNet (Curth & van der Schaar, 2021a) | .20±.18 | .22±.11 | .0200±.0070 | .0200±.0070 | .06±.00 | .02±.00 |
| SNet (Curth & van der Schaar, 2021a) | .09±.10 | .14±.12 | .0040±.0030 | .0040±.0030 | .06±.00 | .02±.00 |
| **NESTER - NEAR** | **.05±.04** | **.05±.03** | **.0034±.0005** | .0039±.0006 | .06±.00 | .02±.01 |
| **NESTER - dPads** | .05±.01 | .05±.02 | .0043±.0001 | **.0028±.0001** | .06±.00 | **.01±.01** |

## 6 Experiments and Results

We perform a comprehensive suite of experiments to study the usefulness of NESTER in estimating treatment effects with our proposed DSL. Our code and instructions to reproduce the results are included in the supplementary material and will be made publicly available.

**Datasets:** Evaluating treatment effect estimation methods requires all potential outcomes to be available (Defn 3.2 and Defn 3.4), which is not possible due to *the fundamental problem of causal inference.* Thus, following Shalit et al. (2017); Yoon et al. (2018); Shi et al. (2019); Farajtabar et al. (2020), we experiment on two semi-synthetic datasets–Twins (Almond et al., 2005), IHDP (Hill, 2011)–that are derived from real-world RCTs (see Appendix § B

Table 3: Results on IHDP, Twins dataset on $\epsilon_{PEHE}$

| Datasets (Metric) → | IHDP ($\sqrt{\epsilon_{PEHE}}(\downarrow)$) | | Twins ($\sqrt{\epsilon_{PEHE}}(\downarrow)$) | |
|---|---|---|---|---|
| Methods ↓ | In-Sample | Out-Sample | In-Sample | Out-Sample |
| OLS-1 | 5.80±0.30 | 5.80±0.30 | .319±.001 | .318±.007 |
| OLS-2 | 2.50±0.10 | 2.50±0.10 | .320±.002 | .320±.003 |
| k-NN | 2.10±0.10 | 4.10±0.20 | .333±.001 | .345±.007 |
| BLR | 5.80±0.30 | 5.80±0.30 | .312±.003 | .323±.018 |
| BART | 2.10±0.10 | 2.30±0.10 | .347±.009 | .338±.016 |
| R Forest | 4.20±0.20 | 6.60±0.30 | .306±.002 | .321±.005 |
| C Forest | 3.80±0.20 | 3.80±0.20 | .366±.003 | .316±.011 |
| BNN | 2.20±0.10 | 2.10±0.10 | .325±.003 | .321±.018 |
| TARNet | 0.88±0.02 | 0.95±0.02 | .317±.005 | .315±.003 |
| MHNET | 1.54±0.70 | 1.89±0.52 | .319±.000 | .321±.000 |
| GANITE | 1.90±0.40 | 2.40±0.40 | **.289±.005** | **.297±.016** |
| CFR$_{WASS}$ | 0.71±0.02 | **0.76±0.02** | .315±.007 | .313±.003 |
| Dragonnet | 1.37±1.57 | 1.42±1.67 | .319±.000 | .321±.000 |
| CMGP | **0.65±0.44** | 0.77±0.11 | .320±.002 | .319±.008 |
| TNet | 0.90±0.01 | 0.91±0.03 | .318±.002 | .319±.000 |
| SNet | 0.69±0.01 | **0.76±0.01** | .318±.002 | .318±.000 |
| **NESTER - NEAR** | 0.73±0.19 | **0.76±0.20** | .318±.002 | .319±.000 |
| **NESTER - dPads** | 0.71±0.10 | **0.76±0.32** | .314±.001 | .331±.001 |

for details). For these two datasets, ground truth potential outcomes (a.k.a. counterfactual outcomes) are synthesized and available, and hence can be used to study the effectiveness of models in predicting potential outcomes. We also experiment on one real-world dataset–Jobs (LaLonde, 1986)–where we observe only one potential outcome. We note that we are commensurate or better than existing work on the number of datasets studied. More details of datasets are provided in Appendix§ B.

**Baselines:** We compare NESTER with different methods including: Ordinary Least Squares with treatment as a feature (OLS-1), OLS with two regressors for two treatments (OLS-2), $k$-Nearest Neighbors ($k$-NN), balancing linear regression (BLR) (Johansson et al., 2016), Bayesian additive regression trees (BART) (Chipman et al., 2010), random forest (Breiman, 2001), causal forest (Wager & Athey, 2018), balancing neural network (BNN) (Johansson et al., 2016), TARNet (Shalit et al., 2017), multi-head network (MHNET) (Farajtabar et al., 2020), Generative Adversarial Nets for inference of individualized treatment effects (GANITE) (Yoon et al., 2018), counterfactual regression with Wasserstein distance (CFR$_{WASS}$) (Shalit et al., 2017), Dragonnet (Shi

et al., 2019), multi-task Gaussian process (CMGP) (Alaa & van der Schaar, 2017) and TNet/SNet (Curth & van der Schaar, 2021a). We implement NESTER using NEAR (Shah et al., 2020) (NESTER-NEAR) that uses neural networks as relaxations of partial programs and dPads (Cui & Zhu, 2021) (NESTER-dPads) that avoids combinatorial search over possible programs using a differentiable pruning strategy.

**Evaluation Metrics:** For the experiments on IHDP and Twins datasets where we have access to both potential outcomes, following (Shalit et al., 2017; Yoon et al., 2018; Shi et al., 2019; Farajtabar et al., 2020), we use the evaluation metrics: $\epsilon_{ATE}$ and $\epsilon_{PEHE}$ (Defn 3.2 and Defn 3.4). $\epsilon_{ATE}$ measures the error in the estimation of the average treatment effect in a population. $\epsilon_{PEHE}$ is operated on the error in the estimation of individual treatment effects. For the experiments on the Jobs dataset where we observe only one potential outcome per data point, following (Shalit et al., 2017; Yoon et al., 2018; Shi et al., 2019; Farajtabar et al., 2020), we use the metric *error in the estimation of average treatment effect on the treated* ($\epsilon_{ATT}$). Definitions and more details of these metrics are provided in Appendix § B. We report both in-sample and out-of-sample performance w.r.t. $\epsilon_{ATE}, \epsilon_{ATT}, \sqrt{\epsilon_{PEHE}}$ in our results. Unlike traditional supervised learning, in-sample performance is non-trivial in this context, since we do not observe counterfactual outcomes (all potential outcomes) during training. Additional details on the experimental setup are presented in Appendix § B.

**Results:** Tabs 4, 3 presents our main results. To permit efficient learning (and to some degree, interpretability of the learned program, as discussed in ablation studies (§ 7), and in Appendix § E), we limit the program depth to utmost 5 for the main experiments. We present results with other depths in ablation studies (§ 7). The results show the superior performance of NESTER over existing methods.

# 7 Additional Empirical Analysis and Discussion

In this section, we present ablation studies to understand various aspects of synthesized programs.

**Flexibility of NESTER:** We study different programs generated by NESTER to estimate treatment effects for different datasets. NESTER has the flexibility to learn both complex models that are required for datasets such as IHDP (complex models such as CMGP outperform simpler models such as OLS on IHDP; see Tab 4) and to learn simple models for datasets such as Twins and Jobs (OLS, $k-$NN perform better on Twins, Jobs compared to complex models). Tab 5 shows sample programs learned by NESTER to estimate treatment effects for various datasets. $\{0..|\mathbf{v}|\}$ is the set of natural numbers from 0 to $|\mathbf{v}|$ (length/size of $\mathbf{v}$). To explain the results further, for each dataset, NESTER has the flexibility to: (i) choose or not choose a specific program primitive; (ii) decide the order in which program primitives are used; and (iii) use a specific program primitive zero or more times. Unlike traditional fixed architectures, this flexibility allows NESTER to use primitives differently for different datasets to perform better.

Table 5: Programs synthesized by NESTER. $|\mathbf{v}|$ = size of $\mathbf{v}$.

| IHDP |
|---|
| if subset($\mathbf{v}, \{0..|\mathbf{v}|\}$) then transform($\mathbf{v}$) else transform($\mathbf{v}$) |
| Twins |
| subset($\mathbf{v}, \{0..|\mathbf{v}|\}$)) |
| Jobs |
| if subset($\mathbf{v}, \{0..|\mathbf{v}|\}$) then  subset($\mathbf{v}, \{0..|\mathbf{v}|\}$) else  subset($\mathbf{v}, \{0..|\mathbf{v}|\}$) |

**Interpretability of NESTER:** We now provide the interpretability to the program generated by NESTER for estimating treatment effects for the Twins dataset: "subset($\mathbf{v}, \{0..|\mathbf{v}|\}$)" (Tab 5). Since the subset primitive allows us to check the performance w.r.t. different subsets of covariates, we empirically verified the effect of choosing a subset of input covariates (other covariates are set to 0) on $\epsilon_{ATE}$. Results in Fig 3 show the performance of NESTER as the number of covariates are increased from 1 to 31 (starting with treatment variable, adding one covariate at a time; we chose this ordering because checking with all possible subsets, $2^{31}$ in this case, is not feasible). This analysis allows us to understand which features are useful to better estimate treatment effects and which are not. For example, if the exclusion of a set of features $S$ is improving a model's performance in estimating treatment effects, $S$ can then be used to infer the causal relationships in the underlying causal structure of the data (e.g., if the sample size is small, not controlling the parents of treatment variable that are not the ancestors of the target variable is likely to improve the precision in estimation of treatment effects Cinelli et al. (2022)). Also, this

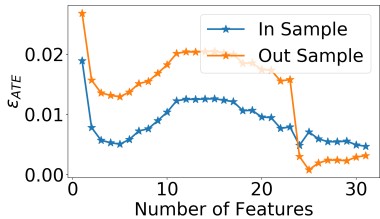

Figure 3: Feature count vs $\epsilon_{ATE}$

simple program synthesized by NESTER supports the fact that simpler models perform better on the Twins dataset. This can be observed from the first three rows and the final row of Tab 4. See Appendix § E for real-world examples of the interpretability of synthesized programs.

**Runtime Analysis:** We also compare the run time of NESTER against the state-of-the-art learning-based method SNet (Curth & van der Schaar, 2021a). As shown in Tab 6, on Twins and Jobs datasets, NESTER-dPads require less time compared to SNet and NESTER-NEAR

Table 6: Run time in minutes

| Dataset | SNet | NESTER-NEAR | NESTER-dPads |
|---------|------|-------------|--------------|
| Twins | 1.85±0.3 | 2.12±0.12 | **1.40±0.20** |
| Jobs | 1.23±0.2 | 1.09±0.40 | **1.00±0.10** |

on average. NESTER-NEAR also achieves state-of-the-art performance with smaller program depths, avoiding heavy computation requirements in practice. Experiments are conducted on a computing unit with an NVIDIA GeForce 1080Ti, and the average time over five runs is reported.

**Choice of DSL:** The choice of DSL impacts the performance of any program synthesis method. We argue that the success of NESTER is because of the specific program primitives in the proposed DSL and their connection to the causal inference literature (Tab 2). Specifically, we study the usefulness of the primitives if − then − else, transform, subset. We conduct an ablation study where the DSL only contains the subset of primitives from the set of primitives 1-4 in the original DSL (Tab 1). When we remove one of the primitives 1-3 from the DSL, we observe the degradation in the performance (Tab 7). Results improved when we added all primitives 1-3 in the DSL.

Table 7: Results on Twins. Primitives 1-4 alone in our proposed DSL are achieving better results compared to the primitives 4-5.

| Metrics → | $\sqrt{\epsilon_{PEHE}}(\downarrow)$ | | $\epsilon_{ATE}(\downarrow)$ | |
|-----------|-----------|-----------|-----------|-----------|
| Primitives of DSL 1 | In-Sample | Out-of Sample | In-Sample | Out-of Sample |
| 1,2,3 | **.318±.003** | **.319±.000** | .0050±.0030 | **.0039±.0006** |
| 1,2,4 | .332±.001 | **.319±.002** | .0210±.0030 | .0140±.0000 |
| 1,3,4 | .332±.001 | **.319±.002** | .0210±.0030 | .0140±.0000 |
| 2,3,4 | .325±.001 | .331±.001 | .0160±.0030 | .0170±.0000 |
| 1,2,3,4 | **.318±.002** | **.319±.000** | **.0034±.0026** | **.0039±.0006** |

**Analysis on Depth of Synthesized Program Structures:** We study the effect of program depth on the estimated treatment effects while keeping all other hyperparameters fixed. Fig 4 shows the results on IHDP and Jobs datasets for various values of program depth. Since IHDP dataset contains 1000 realizations of simulated outcomes (Hill, 2011), we take the first realization and verify the effect of program depth on $\epsilon_{ATE}$. For a program depth of 4,

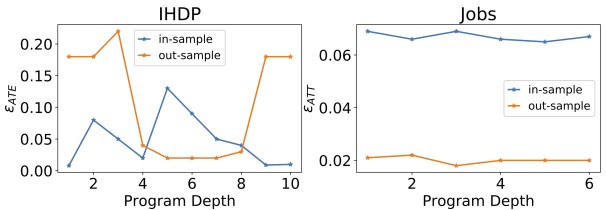

Figure 4: Program depth vs performance.

we observed that the sum of in-sample and out-sample $\epsilon_{ATE}$ is less compared to other depths. We believe that this is because of model over-fitting for large program depths (In Fig 4 left, out-sample $\epsilon_{ATE}$ is increasing while in-sample $\epsilon_{ATE}$ is decreasing). In the Jobs dataset, we observed that almost all program depths results in similar in-sample and out-sample $\epsilon_{ATT}$. Hence, in this case it is advisable to limit the program depth to be a small number as it helps to interpret the results better. On Twins dataset, we observed that simple models give best results. It is observed that, even though we set the hyperparameter that controls the depth of the program graph to be a large value, the resultant optimal program always ends up to be of depth 1, again supporting our claim that simple models work better for the Twins dataset.

## 8 Conclusions

This paper presents an adaptive method for estimating treatment effects using neurosymbolic program synthesis. We propose a domain-specific language for treatment effect estimation by establishing an analogy between parameterized program primitives and model building blocks. The viability and suitability of this approach are theoretically demonstrated. Our approach is validated through extensive experimentation on benchmark datasets, with multiple baselines, showcasing its usefulness. Exploring new program primitives for treatment effect estimation is a promising future direction. NESTER-NEAR encounters time complexity issues for program depths greater than five, but achieves state-of-the-art performance with small program depths. NESTER enables treatment effect estimation from observational data and is particularly well-suited for scenarios with shallow program synthesis requirements. This work has no known detrimental effects.

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

## Appendix

In this appendix, we include the following additional details.

- Proofs of propositions
- Additional details on the experimental setup, including:
  - Details on evaluation metrics
  - Details on datasets
- Example of program synthesis application - FlashFill
- Example of a neurosymbolic program - solving XOR problem
- Interpretability of Synthesized Programs - A real-world example

## A   Proofs of Propositions

**Lemma 5.1.** *(Neural Admissible Relaxations (Shah et al., 2020))* *In an informed search algorithm* $\mathcal{A}$, *given an internal node* $u_i$ *and a leaf node* $u_l$, *let the cost of the leaf edge* $(u_i, u_l)$ *be* $s(r) + \zeta(\mathcal{P}, \theta^*)$, *where* $\theta^* = \arg\min_\theta \zeta(\mathcal{P}, \theta)$ *and* $s(r)$ *is the structural cost in using rule* $r$ *to create* $u_l$ *from* $u_i$. *If a neural network model* $\mathcal{N}$ *with adequate capacity in terms of depth and width of* $\mathcal{N}$ *is used to substitute each non-terminal of* $u_i$, *the training loss of the program obtained is an* $\epsilon-$*admissible heuristic for* $u_i$.

*Proof.* Let $\mathcal{G}$ denote the program graph that is being generated by an informed search algorithm. At any node $u$ in $\mathcal{G}$, let $s(u)$ be the structural cost of $u$ i.e., the sum of costs of rules used to construct $u$. Now, let $u[\alpha_1, \ldots, \alpha_k]$ be any structure (that is not partial) obtained from $u$ by using the rules $\alpha_1, \ldots, \alpha_k$. Then the cost to reach goal node from $u$ is given by:

$$J(u) = \min_{\alpha_1, \ldots, \alpha_n, \theta(u), \theta} [s(u(\alpha_1, \ldots, \alpha_k)) - s(u) + \zeta(u[\alpha_1, \ldots, \alpha_k], (\theta_u, \theta))]$$

where $\theta(u)$ is the set of parameters of $u$ and $\theta$ is the set of parameters of $\alpha_1, \ldots, \alpha_k$. Now, let the heuristic function value $h(u)$ at $u$ be obtained as follows: substitute the non-terminals in $u$ with neural networks parametrized by the set of parameters $\omega$ (these networks are type-correct— for example, if a non-terminal is supposed to generate sub-expressions whose inputs are sequences, then the neural network used in its place is recurrent). Now, let us denote the program obtained by this construction with $(\mathcal{P}(u), (\theta(u), \omega))$. The heuristic function value at $u$ is now given by:

$$h(u) = \min_{\theta(u), \omega} \zeta(\mathcal{P}(u), (\theta(u), \omega)) \tag{6}$$

In practice, neural networks may only form an approximate relaxation of the space of completions and parameters of architectures; also, the training of these networks may not reach global optima. To account for these issues, consider an approximate notion of admissibility Harris (1974); Pearl (1984). For a fixed constant $\epsilon > 0$, let an $\epsilon$-admissible heuristic be a function $h^*(u)$ over architectures such that $h^*(u) \leq J(u) + \epsilon; \forall u$.

As neural networks with adequate capacity are universal function approximators, there exist parameters $\omega^*$ for our neurosymbolic program such that for all $u, \alpha_1, \ldots, \alpha_k, \theta(u), \theta$:

$$\zeta(P(u), (\theta(u), \omega^*)) \leq \zeta(P(u[\alpha_1, \ldots, \alpha_k]), (\theta(u), \theta)) + \epsilon$$

If $s(r) > 0; \forall r \in \mathcal{L}$ (where $\mathcal{L}$ is the DSL under consideration), then $s(u) \leq s(u[\alpha_1, \ldots, \alpha_k])$, which implies:

$$h(u) \leq \min_{\alpha_1, \ldots, \alpha_n, \theta(u), \theta} \zeta(u[\alpha_1, \ldots, \alpha_k], (\theta_u, \theta))) + \epsilon$$
$$\leq \min_{\alpha_1, \ldots, \alpha_n, \theta(u), \theta} \zeta(u[\alpha_1, \ldots, \alpha_k], (\theta_u, \theta))) + s(u(\alpha_1, \ldots, \alpha_k)) - s(u) + \epsilon$$
$$= J(u) + \epsilon$$

In other words, $h(u)$ is $\epsilon$-admissible.

Let $C$ denote the optimal path cost in $\mathcal{G}$. If an informed search algorithm returns a node $u_g$ as the goal node that does not have the optimal path cost $C$, then there must exist a node $u'$ on the frontier (nodes to explore) that lies along the optimal path but has not yet explored. Let $g(u_g)$ denote the path cost at $u_g$ (note that path cost includes the prediction error of the program at $u_g$). This lets us establish an upper bound on the path cost of $u_g$.

$$g(u_g) \leq g(u') + h(u') \leq g(u') + J(u') + \epsilon \leq C + \epsilon.$$

In an informed search algorithm, the heuristic estimate at the goal node $h(u_g)$ is 0. That is, the path cost of the optimal program returned by the informed search algorithm is at most an additive constant $\epsilon$ away from the path cost of the optimal solution. $\qquad\square$

**Proposition 5.1.** *(Universal Approximation Result for NPS)* *If the interventional effect of the treatment variable $T$ on the target variable $Y$ is a continuous function $g(T)$ i.e., $\mathbb{E}[Y|do(T)] = g(T)$, using any DSL $\mathcal{L}$ for synthesizing a single-hidden-layer neural network, NESTER synthesizes a program $(\mathcal{P}, \theta)$ that $\epsilon-$approximates $g$ for a given $\epsilon > 0$.*

*Proof.* We know by universal approximation theorem Hornik et al. (1989) that there exist a trained 1-hidden layer neural network model $\mathcal{N}$ with $d$ inputs $x_1, \ldots, x_d$, $n$ hidden neurons $h_1, \ldots, h_n$, and output $y$ that $\hat{\epsilon}$-approximates $g : \mathbb{R}^d \to \mathbb{R}$ for some $\hat{\epsilon} > 0$. We now show that $\mathcal{N}$'s output can be $\epsilon'$-approximated using a program synthesized using NPS with $\epsilon'$-admissible heuristic and a DSL.

In $\mathcal{N}$, let the activation function used in hidden and output layers be $f(\cdot)$; $\theta_{ij}$ be the weight connecting $i^{th}$ input to $j^{th}$ hidden neuron; and $\theta_j$ be the weight connecting $j^{th}$ hidden neuron to output $y$. The output $y$ of $\mathcal{N}$ can be expressed in terms of inputs, activation function, and parameters as:

$$y = f(\theta_1 f(\theta_{11} x_1 + \cdots + \theta_{d1} x_d) + \cdots + \theta_n f(\theta_{1n} x_1 + \cdots + \theta_{dn} x_d)) \qquad (7)$$

Since the above expression consists of additions, multiplications, and an activation function $f$, it is easy to see that Eqn 7 can be synthesized using the following DSL $\mathcal{L}$:

$$\alpha := \mathtt{f}(\alpha) \mid \mathtt{mul}(\theta, \alpha) \mid \mathtt{add}(\alpha, \alpha) \mid \mathtt{x_1} \mid \ldots \mid \mathtt{x_n}$$

where $\mathtt{mul}, \mathtt{add}$ represent usual multiplication and addition operations. If $d = 2$ and $n = 2$, the synthesized program that matches the expression for $y$ in Eqn 7 looks like:

$$\mathtt{f\ (add\ (mul\ (\theta,\ f(\ add\ (mul\ (\theta,\ x_1), mul(\theta,\ x_2)))), mul(\theta,\ f\ (add\ (mul(\theta,\ x_1), mul(\theta,\ x_2)))))))} \qquad (8)$$

Note that $\theta$ is overloaded in the above expression only for convenience and readability; each $\theta$ is however updated independently while training the above program using gradient descent.

Using Expression 8, it is clear that Eqn 7 can be synthesized using $\mathcal{L}$ for any given $m, n$. Now, as part of our construction, set $s(r) = 0; \forall r \in \mathcal{L}$ to synthesize programs of arbitrary depth and width without worrying about the structural cost of the synthesized program. Now the path cost $p$ of a node $u$ returned by the synthesizer contains only the prediction error value of the program at the node $u$ (Eqn 5). Using Lemma 5.1, $p$ is at most $\epsilon'$ away from the path cost of the optimal solution (node with the expression for $y$, the output of $\mathcal{N}$). Since the path cost of any node only contains the prediction error values, we conclude that the loss incurred by the synthesized program is $\epsilon'-$close to the loss incurred by $\mathcal{N}$.

Finally, as per the universal approximation theorem, we can increase the number of hidden layer neurons of a 1-hidden layer NN $\mathcal{N}$ to approximate $g : \mathbb{R}^d \to \mathbb{R}$ with a certain error, say $\hat{\epsilon}$. Also, there exists a neurosymbolic program $(\mathcal{P}, \theta)$ whose error in approximating $\mathcal{N}$ is $\epsilon'$. Equivalently, there exists a neurosymbolic program $(\mathcal{P}, \theta)$ whose error in approximating $f$ is $(\hat{\epsilon} + \epsilon')$. If we choose $\hat{\epsilon}, \epsilon'$ such that $\epsilon = \hat{\epsilon} + \epsilon'$ for a given $\epsilon$, we have the desired result. $\qquad\square$

Before proceeding with the proof of next proposition, we describe how NESTER, using the proposed DSL can generate programs $(\mathcal{P}_T, \theta_T)$, $(\mathcal{P}_C, \theta_C)$, and $(\mathcal{P}_D, \theta_D)$, whose architectures are the same as TARNet, CFR Shalit et al. (2017), and Dragonnet Shi et al. (2019) respectively.

Table 8: Programs generated using our DSL (Tab 1) equivalent to TARNet ($\mathcal{P}_T$), CFR ($\mathcal{P}_C$), and Dragonnet ($\mathcal{P}_D$).

| $\mathcal{P}_T$ | $\mathcal{P}_C$ | $\mathcal{P}_D$ |
|---|---|---|
| if subset$(\mathbf{v}, \{0\})$ | if subset$(\mathbf{v}, \{0\})$ | if subset$(\mathbf{v}, \{0\})$ |
| then subset$(\mathbf{v}, \{0..|\mathbf{v}|\})$ | then transform$(\mathbf{v})$ | then subset$(\mathbf{v}, \mathtt{pa_T})$ |
| else subset$(\mathbf{v}, \{0, |\mathbf{v}|\})$ | else transform$(\mathbf{v})$ | else subset$(\mathbf{v}, \mathtt{pa_T})$ |

We construct the program architectures $\mathcal{P}_T, \mathcal{P}_C$, and $\mathcal{P}_D$ using the DSL 1. The parameter sets $\theta_T, \theta_C$, and $\theta_D$ are implicit in the program primitives used in the respective architectures. The program architectures for $\mathcal{P}_T, \mathcal{P}_C$, and $\mathcal{P}_D$ are shown in Tab 8. In Tab 8, $\mathtt{pa_T}$ denotes the indices of the parents of $T$ in $\mathbf{v}$.

**Construction of $\mathcal{P}_T$:** TARNet is a simple 2-head network without any constraints on the learned representation $\phi$ (Fig 1). Since there are no constraints on $\phi$, $\mathcal{P}_T$ has two `subset` primitives responsible for learning two representations $\phi_0, \phi_1$ for $p(\mathbf{x}|t = 0)$ and $p(\mathbf{x}|t = 1)$ respectively before producing the estimated potential outcomes (i.e., the outputs of these two `subset` primitives act as the two hypothesis functions $h_0, h_1$ in TARNet to predict the treatment-specific effects.) The condition check for deciding which head to execute is done using `subset`$(\mathbf{v}, \{0\}])$ where `subset` primitive chooses the first index of input and returns $t$ value as its output.

**Construction of $\mathcal{P}_C$:** CFR minimizes the distance between $\phi_0, \phi_1$ (equivalently between $p(\mathbf{x}|t = 0), p(\mathbf{x}|t = 1)$) to achieve IPM regularization. To get similar behavior, $\mathcal{P}_C$ uses `transform` primitive that implicitly generates representations close to each other for inputs with different treatment values. Now, similar to $\mathcal{P}_T$, $\mathcal{P}_C$ has two heads corresponding to two `transform` primitives that output treatment-specific effects.

**Construction of $\mathcal{P}_D$:** In Dragonnet Shi et al. (2019), along with two treatment-specific heads (similar to TARNet), there is another head that predicts the treatment variable so that the parents of the treatment variable are being used for propensity score matching. To achieve this behavior, $\mathcal{P}_D$ uses `subset` primitive that selects parents of the treatment variable $\mathtt{pa_T}$. Once the parent set is chosen, similar to $\mathcal{P}_T$ and $\mathcal{P}_C$, the outputs of two `subset` primitives of $\mathcal{P}_D$ act as the two hypothesis functions $h_0, h_1$ to predict the treatment-specific effects.

**Proposition 5.2.** *(Error Bounds of NESTER) The program $(\mathcal{P}_C, \theta_C)$ generated by NESTER using the proposed DSL, whose architecture is the same as CFR (Shalit et al., 2017), has the same error bounds in estimating treatment effects as that of CFR.*

*Proof.* Since CFR provides error bounds in estimating $\epsilon_{PEHE}$, we show how such bounds can be extended to NESTER. We first restate the following definitions and notations from Shalit et al. (2017).

Let $p^{t=1}(\mathbf{x}) = p(\mathbf{x}|t = 1)$, and $p^{t=0}(\mathbf{x}) = p(x|t = 0)$ denote respectively the treatment and control distributions. Let $\phi : \mathbf{X} \to \mathcal{R}$ be the representation function which is assumed to be one-to-one and differentiable. Let $p_\phi^{t=1}(\mathbf{x}) = p_\phi(\mathbf{x}|t = 1)$, and $p_\phi^{t=0}(\mathbf{x}) = p_\phi(\mathbf{x}|t = 0)$ denote respectively the treatment and control distributions induced over $\mathcal{R}$. Let $h : \mathcal{R} \times \{0, 1\} \to Y$ be a hypothesis function (e.g., treatment-specific heads of TARNet/CFR). The expected loss for the unit $(\mathbf{x}, t)$ is defined as follows

$$l_{h,\phi}(\mathbf{x}, t) \coloneqq \int_Y L(Y^t, h(\phi(\mathbf{x}), t))p(Y^t|\mathbf{x})dY^t$$

Where $L : Y \times Y \to \mathcal{R}^+$ is squared loss function defined as $L(y, \hat{y}) \coloneqq (y - \hat{y})^2$. Now consider the two complimentary loss functions: one is the standard machine learning loss, call the factual loss, denoted by $\epsilon_F$. The other is the expected loss with respect to the distribution where the treatment assignment is flipped, called the counterfactual loss, $\epsilon_{CF}$. These are defined as follows

$$\epsilon_F(h, \phi) \coloneqq \int_{\mathbf{X} \times \{0,1\}} l_{h,\phi}(\mathbf{x}, t)p(\mathbf{x}, t)d\mathbf{x}dt \qquad \epsilon_{CF}(h, \phi) \coloneqq \int_{\mathbf{X} \times \{0,1\}} l_{h,\phi}(\mathbf{x}, t)p(\mathbf{x}, 1 - t)d\mathbf{x}dt$$

Similarly, one can define the expected treated and control losses as follows

$$\epsilon_F^{t=1}(h,\phi) = \int_{\mathbf{X}} l_{h,\phi}(\mathbf{x},1)p^{t=1}(\mathbf{x})d\mathbf{x} \qquad\qquad \epsilon_F^{t=0}(h,\phi) = \int_{\mathbf{X}} l_{h,\phi}(\mathbf{x},0)p^{t=0}(\mathbf{x})d\mathbf{x}$$

$$\epsilon_{CF}^{t=1}(h,\phi) = \int_{\mathbf{X}} l_{h,\phi}(\mathbf{x},1)p^{t=0}(\mathbf{x})d\mathbf{x} \qquad\qquad \epsilon_{CF}^{t=0}(h,\phi) = \int_{\mathbf{X}} l_{h,\phi}(\mathbf{x},0)p^{t=1}(\mathbf{x})d\mathbf{x}$$

For $u := p(t = 1)$, we have the following Shalit et al. (2017)

$$\epsilon_F(h,\phi) = u\epsilon_F^{t=1}(h,\phi) + (1-u)\epsilon_F^{t=0}(h,\phi)$$
$$\epsilon_{CF}(h,\phi) = (1-u)\epsilon_{CF}^{t=1}(h,\phi) + u\epsilon_{CF}^{t=0}(h,\phi)$$

Let $G$ be a function family consisting of functions $g : \mathcal{S} \to \mathbb{R}$. For a pair of distributions $p_1, p_2$ over $\mathcal{S}$, the Integral Probability Metric is defined as follows

$$IPM_G(p_1,p_2) = \sup_{g \in G}\left| \int_{\mathcal{S}} g(s)(p_1(s) - p_2(s))ds \right|$$

For $t \in \{0,1\}$, let $m_t(\mathbf{x}) = E[Y^t|\mathbf{x}]$, $\tau(\mathbf{x}) = m_1(\mathbf{x}) - m_0(\mathbf{x})$ and $\hat{\tau}(\mathbf{x}) = f(\mathbf{x},1) - f(\mathbf{x},0)$ ($f$ is defined in Sec. 3). Then we have the following

$$\epsilon_{PEHE}(f) := \int_{\mathbf{X}} (\hat{\tau}(\mathbf{x}) - \tau(\mathbf{x}))^2 p(\mathbf{x})d\mathbf{x}$$

Let

$$\sigma_{Y^t}^2(p(\mathbf{x},t)) = \int_{\mathbf{X}\times Y} (Y^t - m_t(\mathbf{x}))^2 p(Y^t|\mathbf{x})p(v,t)dY^t d\mathbf{x}$$

and $\sigma_{Y^t}^2 = min\{\sigma_{Y^t}^2(p(\mathbf{x},t)), \sigma_{Y^t}^2(p(\mathbf{x},1-t))\}$

and $\sigma_Y^2 = min\{\sigma_{Y^0}^2, \sigma_{Y^1}^2\}$

Now assume there exists a constant $B_\phi$ and loss $L(y_1, y_2) = (y_1 - y_2)^2$ such that for $t \in \{0,1\}$, the functions $g_{\phi,h}(r,t) := \frac{1}{B_\phi}l_{h,\phi}(\psi(r),t) \in G$. Then we have

$$\epsilon_{PEHE}(h,\phi) \le 2(\epsilon_{CF}(h,\phi) + \epsilon_F(h,\phi) - 2\sigma_Y^2) \le 2(\epsilon_F^{t=0}(h,\phi) + \epsilon_F^{t=1}(h,\phi) + B_\phi IPM_G(p_\phi^{t=0}, p_\phi^{t=1}) - 2\sigma_Y^2) \quad (9)$$

We refer to Shalit et al. (2017) for the complete proof of the inequality 9 which is valid for Counterfactual Regression (CFR) model Shalit et al. (2017). We now present the following equivalences to show that the above error bound is valid for the program $(\mathcal{P}_C, \theta_C)$ equivalent to CFR.

- In $\mathcal{P}_C$, `subset(v, {0})` acts as the decision node to decide which specific `transform(v)` to execute. The outputs of these specific `transform` are the same as the outputs of the hypothesis function $h$ used in the factual and counterfactual losses $\epsilon_F, \epsilon_{CF}$ defined earlier.

- By our construction of $(\mathcal{P}_C, \theta_C)$, we have a two `transform` primitives to output $p_\phi^{t=0}$ and $p_\phi^{t=1}$. $\phi$ is trained to minimize the MMD between $p_\phi^{t=0}$ and $p_\phi^{t=1}$. Since MMD is one specific IPM, we replace IPM with MMD in the inequality 9.

- $\sigma_Y^2$ can be directly obtained from the observational data. Hence the error bounds guaranteed by NESTER w.r.t. $\epsilon_{PEHE}$ is as follows.

$$\epsilon_{PEHE}(h,\phi) \le 2(\epsilon_{CF}(h,\phi) + \epsilon_F(h,\phi) - 2\sigma_Y^2) \le 2(\epsilon_F^{t=0}(h,\phi) + \epsilon_F^{t=1}(h,\phi) + B_\phi MMD(p_\phi^{t=0}, p_\phi^{t=1}) - 2\sigma_Y^2)$$

$\square$

# B Experimental Setup

## B.1 Additional Details on Evaluation Metrics

For the experiments on IHDP and Twins datasets where we have access to both potential outcomes, following Shalit et al. (2017); Yoon et al. (2018); Shi et al. (2019); Farajtabar et al. (2020), we use the evaluation metrics: *error in the estimation of Average Treatment Effect ($\epsilon_{ATE}$)* and the *expected Precision in Estimation of Heterogeneous Effect ($\epsilon_{PEHE}$)*. For a sample of $n$ data points, $\epsilon_{ATE}$, $\epsilon_{PEHE}$ are defined as follows.

$$\epsilon_{ATE} := |\frac{1}{n}\sum_{i=1}^{n}[f(\mathbf{x}_i, 1) - f(\mathbf{x}_i, 0)] - \frac{1}{n}\sum_{i=1}^{n}[Y_i^1 - Y_i^0]|$$

$$\epsilon_{PEHE} := \frac{1}{n}\sum_{i=1}^{n}[(f(\mathbf{x}_i, 1) - f(\mathbf{x}_i, 0)) - (Y_i^1 - Y_i^0)]^2$$

For the experiment on the Jobs dataset where we observe only one potential outcome per data point, following Shalit et al. (2017); Yoon et al. (2018); Shi et al. (2019); Farajtabar et al. (2020), we use the metric *error in estimation of Average Treatment Effect on the Treated* ($\epsilon_{ATT}$), which is defined as follows.

$$\epsilon_{ATT} := |ATT^{true} - \frac{1}{|T|}\sum_{i\in T}[f(\mathbf{x}_i, 1) - f(\mathbf{x}_i, 0)]| \tag{10}$$

where $ATT^{true}$ is defined as:

$$ATT^{true} := \frac{1}{|T|}\sum_{i\in T}Y_i^1 - \frac{1}{|U \cap E|}\sum_{i\in U\cap E}Y_i^0 \tag{11}$$

and $T$ is the treated group, $U$ is the control group, and $E$ is the set of data points from a randomized experiment Shalit et al. (2017) (see description of Jobs dataset below for an example of $E, T$, and $U$).

**Understanding $k$-NN results:** In $k$-NN algorithm, if $k = 1$ and treatment value $t = 1$, $f(\mathbf{x}_i, 1)$ is exactly same as $Y_i^1$. If treatment value $t = 0$, $f(\mathbf{x}_i, 0)$ is exactly same as $Y_i^0$ because of the way k-NN works during test time on in-sample data. For this reason, the estimated value of $\epsilon_{ATE}$ is biased towards 0. This bias exists even for higher values of $k$ in $k$-NN while taking the average outputs of $k$ nearest data points. However, we do not observe such bias w.r.t. out-sample data. Hence, following earlier work Yoon et al. (2018), we only consider K-NN results for out-sample performance.

## B.2 Details on Datasets

**IHDP:** Infant Health and Development Program (IHDP) is a randomized control experiment on 747 low-birth-weight, premature infants. The treatment group consists of 139 children, and the control group has 608 children. The treatment group received additional care such as frequent specialist visits, systematic educational programs, and pediatric follow-up. The Control group only received pediatric follow-up. Hill (2011) created the semi-synthetic version of IHDP dataset by synthesizing both potential outcomes. Following Hill (2011); Shalit et al. (2017); Yoon et al. (2018); Shi et al. (2019), we use simulated outcomes of the IHDP dataset from NPCI package Dorie (2016). This experiment aims to estimate the effect of treatment on the IQ score of children at the age of 3.

**Twins:** The Twins dataset is derived from all births in the USA between 1989-1991 Almond et al. (2005). Considering twin births in this period, for each child, we estimate the effect of birth weight on 1-year mortality rate. Treatment $t = 1$ refers to the heavier twin and $t = 0$ refers to the lighter twin. Following Yoon et al. (2018), for each twin-pair, we consider 30 features relating to the parents, the pregnancy, and the birth. We only consider twins weighing less than 2kg and without missing features. The final dataset has 11,400 pairs of twins whose mortality rate for the lighter twin is 17.7%, and for the heavier 16.1%. In this setting, for each twin pair we observed both the case $t = 0$ (lighter twin) and $t = 1$ (heavier twin) (that is, since all other features such as parent's race, health status, gestation weeks prior to birth, etc. are same except the weight of

Table 9: Dataset details. 'Input Size' includes the treatment variable.

| Dataset | Sample Size | Input Size | Batch Size | Epochs | Train/Valid/Test Split (%) |
|---------|-------------|------------|------------|--------|----------------------------|
| IHDP | 747 (1000 such instances) | 26 | 16 | 100 | 64/16/20 |
| Twins | 11400 | 31 | 128 | 7 | 64/16/20 |
| Jobs | 3212 | 18 | 64 | 10 | 64/16/20 |

Table 10: **Left**: An example FlashFill task where input names are automatically translated to an output format in which last name is followed by the initial of the first name; **Right**: The DSL for FlashFill task based on regular expression string transformations Parisotto et al. (2016).

| Input | Output |
|-------|--------|
| William Henry Charles | Charles, W. |
| Michael Johnson | Johnson, M. |
| Barack Rogers | Rogers, B. |
| Martha D. Saunders | Saunders, M. |
| Peter T Gates | Gates, P. |

$$\texttt{String e} := \texttt{Concat}(\texttt{f}_1, \ldots, \texttt{f}_\texttt{n})$$

$$\texttt{Substring f} := \texttt{ConstStr}(\texttt{s})|\texttt{SubStr}(\texttt{v}, \texttt{p}_\texttt{l}, \texttt{p}_\texttt{r})$$

$$\texttt{Position p} := (\texttt{r}, \texttt{k}, \texttt{dir})|\texttt{ConstPos}(\texttt{k})$$

$$\texttt{Direction Dir} := \texttt{Start}|\texttt{End}$$

$$\texttt{Regex r} := \texttt{s}|\texttt{T}_1|\ldots|\texttt{T}_\texttt{n}$$

each twin, the choice of twin (lighter vs heavier) is associated with the treatment ($t = 0$ *vs* $t = 1$)); thus, the ground truth of individualized treatment effect is known in this dataset. In order to simulate an observational study from these 11,400 pairs, following Yoon et al. (2018), we selectively observe one of the two twins using the feature information $\mathbf{x}$ (to create selection bias) as follows: $t|\mathbf{x} \sim \texttt{Bernoulli}(\texttt{sigmoid}(\mathbf{w}^T\mathbf{x} + n))$ where $\mathbf{w}^T \sim U((-0.1, 0.1)^{30 \times 1})$ and $n \sim N(0, 0.1)$.

**Jobs:** The Jobs dataset is a widely used real-world benchmark dataset in causal inference. In this dataset, the treatment is job training, and the outcomes are income and employment status after job training. The dataset combines a randomized study based on the National Supported Work Program in the USA (we denote the set of observations from this randomized study with $E$) with observational data A. Smith & E. Todd (2005). Each observation contains 18 features such as age, education, previous earnings, etc. Following Shalit et al. (2017); Yoon et al. (2018), we construct a binary classification task, where the goal is to predict unemployment status given a set of features. The Jobs dataset is the union of 722 randomized samples ($t = 1 : 297, t = 0 : 425$) and 2490 observed samples ($t = 1 : 0, t = 0 : 2490$). The treatment variable is job training ($t = 1$ if trained for job else $t = 0$), and the outcomes are income and employment status after job training. In Eqns 10-11, we then have $|T| = 297, |C| = 2915, |E| = 722$. Since all the treated subjects $T$ were part of the original randomized sample E, we can compute the true ATT (Eqn 11) and hence can study the precision in the estimation of $ATT$ (Eqn 10).

Tab 9 summarizes the dataset details. Each dataset is split 64/16/20% into train/validation/test sets, similar to earlier efforts. All experiments were conducted on a computing unit with a single NVIDIA GeForce 1080Ti.

## C   FlashFill Task and Semantics of its DSL

Following our discussion in Section 1, for better understanding of symbolic program synthesis, we provide an example of a symbolic program application called FlashFill Parisotto et al. (2016). Examples of the FlashFill task and a DSL to synthesize programs that solve FlashFill task are given in Tab 10. Semantics of the DSL in Tab 10 Right are as follows.

- $\texttt{Concat}(\texttt{f}_1, \ldots, \texttt{f}_\texttt{n})$ - concatenates the results of the expressions $\texttt{f}_1, \ldots, \texttt{f}_\texttt{n}$.
- $\texttt{ConstStr}(\texttt{s})$ - returns the constant string $\texttt{s}$.
- $\texttt{SubStr}(\texttt{v}, \texttt{p}_\texttt{l}, \texttt{p}_\texttt{r})$ - returns substring $\texttt{v}[\texttt{p}_\texttt{l}..\texttt{p}_\texttt{r}]$ of the string $\texttt{v}$, using position logic corresponding to $\texttt{p}_\texttt{l}, \texttt{p}_\texttt{r}$. $\texttt{v}[\texttt{i}..\texttt{j}]$ denotes the substring of string $\texttt{v}$ starting at index $\texttt{i}$ (inclusive) and ending at index $\texttt{j}$ (exclusive), and $\texttt{len}(\texttt{v})$ denotes the length of the string $\texttt{v}$

Table 11: **Left:** A neurosymbolic program to solve XOR problem. **Right:** Smooth approximation of the program on the left where $\sigma$ is sigmoid function. $\beta$ is a temperature parameter. As $\beta \to 0$, the approximation approaches usual $\mathtt{if - then - else}$ (Section 4.1).

```
if affine[1,1;0](x) > 0 then
    if affine[1,1;-1](x) > 0 then
        affine[0,0;0](x)
    else
        affine[1,1;0](x)
else
    affine[0,0;0](x)
```

$\sigma(\beta \times \mathtt{affine}_{[1,1;0]}(\mathtt{x})) \times$
$(\sigma(\beta \times \mathtt{affine}_{[1,1;-1]}(\mathtt{x})) \times \mathtt{affine}_{[0,0;0]}(\mathtt{x}) +$
$(1 - \sigma(\beta \times \mathtt{affine}_{[1,1;-1]}(\mathtt{x}))) \times \mathtt{affine}_{[1,1;0]}(\mathtt{x})) +$
$(1 - \sigma(\beta \times \mathtt{affine}_{[1,1;0]}(\mathtt{x}))) \times \mathtt{affine}_{[0,0;0]}(\mathtt{x})$

- $\mathtt{ConstPos(k)}$ - returns $\mathtt{k}$ if $\mathtt{k} \geq 0$ else return $\mathtt{l + k}$ where $\mathtt{l}$ is the length of the string
- $(\mathtt{r, k, Start})$ - returns the Start of $\mathtt{k^{th}}$ match of the expression $\mathtt{r}$ in $\mathtt{v}$ from the beginning (if $\mathtt{k} \geq 0$) or from the end (if $\mathtt{k} < 0$).
- $(\mathtt{r, k, End})$ - returns the End of $\mathtt{k^{th}}$ match of the expression $\mathtt{r}$ in $\mathtt{v}$ from the beginning (if $\mathtt{k} \geq 0$) or from the end (if $\mathtt{k} < 0$).

Based on the above semantics, a program that generates the desired output given the input names in Tab 10 is: $\mathtt{Concat(f_1, ConstStr(","), f_2, ConstStr("."))}$ where $\mathtt{f_1 \equiv SubStr(v, (" ", -1, End), ConstPos(-1))}$ and $\mathtt{f_2 \equiv SubStr(v, ConstPos(0), ConstPos(1))}$.

## D Neurosymbolic Program Example: Solving XOR Problem

Following our discussion in Section 3, for better understanding of the internal workings of a neurosymbolic program, we provide an example on solving the XOR problem i.e., predicting the output of XOR operation given two binary digits. Unlike symbolic programs, neurosymbolic programs are differentiable and can be trained using gradient descent. Program primitives in a neurosymbolic program have trainable parameters associated with them. The program shown in Tab 11 (left) is constructed using (i) $\mathtt{if - then - else}$ and (ii) $\mathtt{affine}$ program primitives. $\mathtt{affine}$ primitive takes a vector as input and returns a scalar that is the sum of dot product of parameters with the input and a bias parameter. For example, if $\mathtt{x} = [1, 0]$ then $\mathtt{affine}_{[\theta_1, \theta_2; \theta_3]}(\mathtt{x}) = \theta_1 \times 1 + \theta_2 \times 0 + \theta_3 = \theta_1 + \theta_3$. The subscripts of $\mathtt{affine}$ in $\mathtt{affine}_{[\theta_1, \theta_2; \theta_3]}$ contain the parameters $\theta_1, \theta_2$ and bias parameter $\theta_3$ separated by semi colon (;). The smooth approximation of this program, to enable backpropagation, is shown in Tab 11 (right). The parameter values are hard-coded for illustation purposes. In practice, these weights are learned by training through gradient descent.

## E Interpretability of Synthesized Programs: A Real World Example

We expect that each program primitive in a domain-specific language has a semantic meaning; hence, interpretability in program synthesis refers to understanding the decision of a synthesized program using various aspects such as: which program primitives are used and why? what does the learned sequence of program primitives mean for the problem? what is the effect of each program primitive on the output? etc.

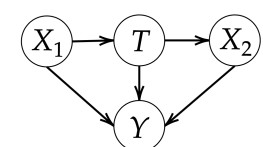

Figure 5: A real-world example for interpreting the synthesized programs.

We explain more clearly with an example. Consider a causal model consisting of variables $T, X_1, X_2, Y$ as shown in Fig 5 where: (i) $X_1$ causes $T$ and $Y$; (ii) $T$ causes $X_2$ and $Y$; and (iii) $X_2$ causes $Y$. A real-world scenario depicted by this causal model could be where $T$ is the *average distance walked by a person in a day*, $X_1$ is *age*, $X_2$ is *metabolism*, and $Y$ is *blood pressure*.

In this example, our goal is to estimate the effect of *walking* ($T$) on *blood pressure* ($Y$). In this case, the ideal estimator for the quantity $\mathbb{E}[Y|do(t)]$ is $\sum_{x_1 \sim X_1} \mathbb{E}[Y|t, x_1]p(x_1)$. However, NESTER has access to only observational data and is unaware of the underlying causal process. Now consider the following two possible programs $p_1, p_2$ synthesized by NESTER to estimate the treatment effect of $T$ on $Y$. Let $\mathbf{v} = [t, x_1, x_2]$ be an input data point.

$$p_1 : \texttt{if subset}(\mathbf{v}, \{0\}]) \qquad\qquad p_2 : \texttt{if subset}(\mathbf{v}, \{0\}])$$
$$\texttt{then subset}(\mathbf{v}, \{0, 1\}]) \qquad\qquad \texttt{then subset}(\mathbf{v}, \{0, 1, 2\})$$
$$\texttt{else subset}(\mathbf{v}, \{0, 1\}]) \qquad\qquad \texttt{else subset}(\mathbf{v}, \{0, 1, 2\})$$

The only difference between $p_1$ and $p_2$ is the set of indices used in $\texttt{subset}$ primitives. $p_1$ uses only $T, X_1$ (indicated by $\{0, 1\}$ in $p_1$) to predict $Y$; while $p_2$ uses $T, X_1, X_2$ (indicated by $\{0, 1, 2\}$ in $p_2$) to predict $Y$. In this case, we would ideally observe $p_1$ to perform better than $p_2$ because $p_1$ controls for the correct set of confounding variables ($\{X_1\}$ in this case). Conversely, observing a strong performance for $p_1$ tells us that $\{X_1\}$ is the confounder, without knowledge of the causal model.

Observing the generated program and primitives gives us insights about the underlying data-generating process such as which features are the potential causes of treatment (e.g., *age* affects the *average distance* a person can walk), which features should not be controlled (e.g., we need the effect of *walking* on *blood pressure* irrespective of the *metabolism* rate of a person), etc. Such information encoded in a synthesized program can also be validated with domain experts if available. Our experimental results and ablation studies discussed above show other ways of interpreting programs.

