# OpenReview forum: "NESTER: An Adaptive Neurosymbolic Method for Treatment Effect Estimation"
_TMLR — Rejected by TMLR_

### Review · Reviewer_T5Zs · 2023-04-23

**Summary Of Contributions:**

This paper proposes a neurosymbolic program synthesis (NPS) method called NESTER for estimating treatment effects. The synthesized programs correspond to model architectures, and the main motivation in my view is to be able to adapt the architecture of neural network-based estimators to different datasets (i.e., data distributions). The main contribution of the authors toward this is a domain-specific language (DSL) of program primitives/architecture components based on NN architectures proposed in the treatment effect estimation literature. The synthesis algorithm appears to be adopted from Shah et al. (2020) and uses A* search. The performance of NESTER in treatment effect estimation is compared to a comprehensive set of baselines on three commonly-used datasets for the task. This is followed by analysis of the synthesized programs/architectures, runtime, and sensitivity to DSL primitives and program depth. Theoretical statements are also given on the suitability of NPS for treatment effect estimation and its ability to reproduce the CFR architecture of Shalit et al. (2017) and associated error bounds.

**Audience:**

Yes

**Claims And Evidence:**

No

**Requested Changes:**

I think the items listed under Weaknesses are all important for me to recommend acceptance.

## Response After Rebuttal
Thanks to the authors for their detailed responses to my comments. I cannot recommend acceptance at this time because the revisions and clarifications are numerous and occur throughout the manuscript. I think they are best left to a full review cycle corresponding to a resubmission.

I think the change that will be most impactful in improving presentation for me (and hopefully others) is to emphasize right when they first appear that the primitives $\texttt{subset}$ and $\texttt{transform}$ (now $\texttt{align}$) include an MLP that maps the resultant vector to a scalar output. I think leaving it until Table 1 is too late to correct the confusion.

**Strengths And Weaknesses:**

## Strengths
- The first neurosymbolic approach to treatment effect estimation (according to the authors), which goes some way toward unifying works in the NN-based treatment effect estimation literature.
- The experimental results show that NESTER indeed adapts the architecture to each dataset (Section 7, Flexibility and Interpretability subsections), which results in superior performance in estimating average treatment effects (ATE, Table 3).
- The set of baselines used in the experiments is comprehensive.

## Weaknesses
Overall, I found numerous issues with clarity and completeness.

- I did not find helpful the example programs shown, especially in Figure 1 in the introduction and also later in Table 4. Perhaps the notation for the program primitives needs to be made more detailed.
    - In Figure 1, variable $v$ has not been defined at this point, not to mention $\texttt{subset(v, \{0\})}$ and how it evaluates to True/False. This only becomes clearer in Section 4.2, paragraph 2, but even there the implicit assumption seems to be that $t$ is binary and $t=1$ corresponds to True and $t=0$ to False.
    - In Table 4, isn't $\texttt{subset}(\mathbf{v}, \{0,\dots,|\mathbf{v}|\})$ an identity operation? More importantly, how does its vector output evaluate to True/False?
    - In both Figure 1 and Table 4, the identical THEN and ELSE clauses are very confusing: it seems (certainly when Figure 1 is introduced) that the IF statement has no effect.
- Section 2 makes more clear that "multi-head NN architectures with regularizers" is arguably the favored class of estimators and the focus of the work. If this is correct, then I think the focus on this class should be made more clear in the Introduction.
- Section 3, Neurosymbolic Program Synthesis subsection: I found this summary hard to follow because of all the terminology that is unfamiliar to me, as someone who knows about treatment effect estimation but not NPS. I think what would help is to make each term/concept concrete using examples from Figure 2 and perhaps additional figures. It was also hard to understand what the structural cost $s(\mathcal{P})$ means concretely.
- It was not clear to me until Section 4.2 that the goal in this paper is to estimate $\mathbb{E}[Y | T=t, X=x]$. I think this should be stated earlier and would explain why $\zeta(\mathcal{P}, \theta)$ in (5) is defined the way it is. Similarly, the first sentence of Section 4.1 states that the paper focuses on "the problem of mapping a set of observational input data points to the corresponding potential outcomes," which I again think would be clearer as *observed* outcomes instead of "potential".
- Section 4.3: This section was the most unclear for me.
    - What is the "goal node" here?
    - What is meant by "non-terminals in an internal node $u$ are substituted by an NN"? What are the inputs and outputs of these NNs? What about non-terminals $\alpha_1$ that are conditions of IF clauses?
    - Why are multiple programs returned? I see Algorithm 1 returning only a single leaf node $v$.
    - What is the cross-validation score? The same as eq. (5)?
    - Is limited depth encouraged by the structural cost $s(\mathcal{P})$?
    - Algorithm 1, line 3: $f(u)$ is not defined until later in line 10.
- Section 5: Vagueness of theoretical statements
    - Lemma 5.1: This lemma partly justifies the use of training loss as a valid heuristic function (underestimator of the true cost). But without specifying the value of $\epsilon$ in terms of problem parameters (perhaps the training set size?), the statement seems empty to me.
    - Proposition 5.1: I find the statement of this result too vague. I think one needs to state for example how the continuous function $g$ relates to what the synthesized programs do. Also, the sentence following Proposition 5.1 could be used to refine the statement beyond "there exists a DSL."
- The PEHE metric is not included in Table 3, and Appendix C only shows PEHE results for the IHDP dataset. I think it is important for completeness to report PEHE on the other datasets too, and ideally in a table in the main paper rather than the appendix since individual treatment effect (ITE) estimation is an important objective.
- I think there should be more discussion of how the synthesized programs, in Table 4 and in general, relate to specific existing NN architectures. Has NESTER come up with anything significantly different in the authors' experience? One particular aspect of this is how the complexity of NPS-generated estimators compares to those in the literature.
- Section 7, Interpretability of NESTER
    - It is not clear how NESTER performs subset selection. I imagine it is not through exhaustive evaluation.
    - In the same vein, the results in Figure 3 do not definitively establish that including all covariates is best, if they were added one at a time in some order. This is only a small fraction of all subsets. Also, the out-sample curve does not have a minimum at all 31 covariates.
- Section 7, Choice of DSL: The ablation study seems incomplete.
    - Why was primitive 4 (arithmetic operation) not considered for ablation?
    - None of the returned programs in Table 4 used primitive 4, so why is the combination 1,2,3 worse than 1,2,3,4?
    - Why is combination 2,3,4 missing?

## Minor comments and questions
- Section 2, paragraph 1
    - "difference in potential outcomes between the treatment and corresponding control data points": Should it be observed outcomes, not potential?
    - "is estimated as treatment effect" --> "is the estimate of treatment effect"?
- Section 2, paragraph 2: Please check the formatting of citations such as L & T, 2013.
- Eq. (2): I believe the square should be inside the expectation.
- Eq. (3): The notation $Y^0$, $Y^1$ has already been defined so I do not see the need to also use Pearl's $do$ notation.
- Section 4.1, "each of these listed primitives outputs a real scalar number": Can the output be a vector, not necessarily scalar?
- Table 1, primitives 2 and 3
    - I wonder why "feed the resultant vector into an MLP" is part of both these primitives and not its own primitive.
    - I think $\texttt{transform}$ is too generic a term for the IPM-regularized embedding that this represents. Perhaps $\texttt{align}$?
- Proposition 5.2: I think the real statement is that CFR is indeed an instance of the programs that can be synthesized? Then the error bounds follow from that fact.
- Section 7, "Interpretability of NESTER": I think this is not so much interpretability as verifying that NESTER's choice of not taking a proper subset of the covariates is correct.
- Section 7, Runtime Analysis: What accounts for the bulk of the complexity of NESTER? Is it training multiple NNs with different architectures considered in the search?
- Section 7, Analysis of Depth: Several things to make more precise.
    - What does "take the first instance" mean?
    - Can the "better trade-off between in-sample and out-sample $\epsilon_{ATE}$" be made precise? Does a depth of 4 minimize the sum of in-sample and out-sample $\epsilon_{ATE}$?
    - In the description of Figure 4 left panel, out-sample $\epsilon_{ATE}$ increases only after depth 6, while in-sample $\epsilon_{ATE}$ does decrease after depth 5 but only after a jump up from depth 4 to 5.

---

> ### Author Response · Authors · 2023-05-24
> **Response to the reviewer T5Zs**
>
> Thank you for your detailed and insightful review. We address each of your concerns below. We've also uploaded the revised version of the manuscript addressing the concerns.
>
> >I did not find helpful the example programs shown, especially in Figure 1 in the introduction and also later in Table 4...
> > * In Figure 1, variable $v$ has not been defined at this point, not to mention $\mathtt{subset(\mathbf{v},\{0\})}$ and how it evaluates to True/False. This only becomes clearer in Section 4.2, paragraph 2, but even there the implicit assumption seems to be that  is binary and  corresponds to True and  to False.
> > * In Table 4, isn't $\mathtt{subset(\mathbf{v},\{0..|v|\})}$ an identity operation? More importantly, how does its vector output evaluate to True/False?
> > * In both Figure 1 and Table 4, the identical THEN and ELSE clauses are very confusing: it seems (certainly when Figure 1 is introduced) that the IF statement has no effect.
>
> * In the introduction, we motivate our method by showing that our method is a generalization of existing multi-head neural network models. We agree that notations have to be introduced upfront to understand the programs well. We've updated the caption of Figure 1 and the paragraph accompanying Figure 1 in the revised version to introduce the notations required to understand the figure.
> * As explained in Section 4.1 and Table 1, $\mathtt{subset(\mathbf{v},\{0..|v|\})}$ returns a scalar value, not a vector. Since we implement a smooth approximation of the $\mathtt{if-then-else}$ primitive (as explained in the first paragraph after Table 1),  $\mathtt{subset(\mathbf{v},\{0..|v|\})}$ doesn't have to be evaluated to exact True/False. The output of $\mathtt{subset(\mathbf{v},\{0..|v|\})}$ primitive is a scalar value and can be used to execute one of the two heads of $\mathtt{if-then-else}$ primitive in a smooth manner so that backpropagation is possible.
> * In the programs shown in Figure 1, the $\mathtt{if}$ primitive is used to resemble the two-head architecture used in literature, and the $\mathtt{then, else}$ clauses are used to mimic the specific regularization terms used in the existing methods. Depending on the condition check in $\mathtt{if}$ clause, respective $\mathtt{then}$ or $\mathtt{else}$ clauses get executed. Consider $\mathcal{P}_C$ of Figure 1 to understand this better. We first note that we pre-train $\mathtt{\phi}$ in $\mathtt{transform}$ primitive to minimize MMD between treatment and control distributions, which is the same as the CFR IPM regularization. During inference, for a given input, the decision of which head (i.e., $\mathtt{then, else}$) to execute depends on $\mathtt{if}$ condition. Even if the primitive $\mathtt{transform}$ in $\mathtt{then, else}$ clauses look similar, they produce different outputs. That is, $\mathtt{transform}$ primitive use a representation $\phi$ from a pre-trained model and then the corresponding MLPs in respective $\mathtt{transform}$ primitives produce different outputs.
>
> > Section 2 makes more clear that "multi-head NN architectures with regularizers" is arguably the favored class of estimators and the focus of the work. If this is correct, then I think the focus on this class should be made more clear in the Introduction.
>
> Yes, our method is motivated by the observation that multi-head NN architectures with regularizers are powerful and can be generalized using the proposed neurosymbolic program synthesis method. In the revised manuscript, we've updated the following lines in the introduction (bold text indicates the added/changed content) to make it clear.
>
> * Paragraph accompanying Figure 1 --> In this paper, we provide a pathway to integrate existing solutions based on the potential outcomes framework, **especially multi-head NN architectures with regularizers**, into a single framework and propose a generalized method for treatment effect estimation. As shown in Fig 1, our method generates programs, each of which can instantiate existing methods **based on multi-head neural network (NN) architectures** for treatment effect estimation as special cases.
> * Figure 1 Caption --> Programs $\mathcal{P}_T, \mathcal{P}_C, \mathcal{P}_D$ generated by the proposed method, NESTER, using our Domain-Specific Language (DSL) (Table 1) that achieve functionality similar to **the popular multi-head NN models** TARNet ($\mathcal{P}_T$), CFR ($\mathcal{P}_C$), and Dragonnet ($\mathcal{P}_D$).
>
> Having said that, it is worth noting that, even though the motivation for our method is multi-head NN architectures and the regularizers therein, our method generates programs that are outside of the multi-head NN architecture family (e.g., a simple program generated for the Twins dataset as shown in Table 4) making our method richer (in terms of programs generated) than just the multi-head NN architecture family, thanks to the adaptability of neurosymbolic program synthesis.

---

> > ### Author Response · Authors · 2023-05-24
> > **Response to reviewer T5Zs (continued)**
> >
> > > Section 3, Neurosymbolic Program Synthesis subsection: I found this summary hard to follow because of all the terminology that is unfamiliar to me, as someone who knows about treatment effect estimation but not NPS....
> >
> > We've extended Figure 2 in the revised version of the paper to explain program synthesis with an example.
> >
> > Structural cost $s(\mathcal{P})$ is the sum of structural costs of rules used to generate a program $\mathcal{P}$ from the root node. We've changed/updated the following line in Section 3 --> Neurosymbolic program Synthesis subsection to explain the definition of the cost of a production rule.
> >
> > *Let $s(r)$ be the cost incurred in using the production rule $r$ for generating a program structure (leaf node) or partial structure (internal node) from a given partial structure (internal node).*
> >
> > We've added the line below to Section 3--> Neurosymbolic program Synthesis subsection to better explain the extended Figure 2.
> >
> > *This paper uses $A$-star informed search algorithm to generate a program tree. While generating a program tree, a node $u$ with minimum $f(u)$ value is expanded next, where $f(u)=s(\mathcal{P}(u))+h(u)$ is the sum of the structural cost $s(\mathcal{P}(u))$ of the partial structure $\mathcal{P}(u)$ in $u$ plus the heuristic value $h(u)$ at the node $u$ that underestimates the cost to reach goal node from $u$.*
> >
> >
> > > It was not clear to me until Section 4.2 that the goal in this paper is to estimate $\mathbb{E}[Y|T=t, \mathbf{X}=\mathbf{x}]$. I think this should be stated earlier and would explain why $\zeta(\mathcal{P},\theta)$ in (5) is defined the way it is.
> >
> > Thank you for pointing this out. In the revised version, we've added the following line to the paragraph after Equation 4 in Section 3.
> >
> > *Hence, this paper aims to synthesize programs that compute the quantity $\mathbb{E}[Y|T=t, \mathbf{X}=\mathbf{x}]$ given the observational data $\mathcal{D}$.*
> >
> > Similarly, we've added the following line to the paragraph before Equation 5 to justify why $\zeta(\mathcal{P},\theta)$ is defined the way it is.
> >
> > *Since the goal of this paper is to synthesize a program $(\mathcal{P}, \theta)$ that estimates the quantity $\mathbb{E}[Y|T=t, \mathbf{X}=\mathbf{x}]$, which can be modeled as a regression problem, the squared error is a good choice for assessing the performance of the program $(\mathcal{P}, \theta)$ in estimating potential outcomes.*
> >
> > Now, from the line before Equation 5 (i.e., The overall goal of NPS is then to find a structurally simple program with low prediction error, i.e., to solve the following optimization problem), our goal is to find the structurally simple program (i.e., minimum $s(\mathcal{P})$) with low prediction error (i.e., $\zeta(\mathcal{P}, \theta)$), justifying the final objective in Equation 5.
> >
> > >The first sentence of Section 4.1 states that the paper focuses on "the problem of mapping a set of observational input data points to the corresponding potential outcomes," which I again think would be clearer as observed outcomes instead of "potential".
> >
> > Thank you for pointing it out. We've changed the first sentence of Section 4.1 as follows (bold text  indicates the updated content)
> >
> > *We pose the problem of treatment effect estimation as the problem of mapping a set of observational input data points to the corresponding **observed** outcomes.*

---

> > > ### Author Response · Authors · 2023-05-24
> > > **Response to reviewer T5Zs (continued)**
> > >
> > > > Section 4.3
> > > > * What is the "goal node" here?
> > > > * What is meant by "non-terminals in an internal node $u$ are substituted by an NN"? What are the inputs and outputs of these NNs? What about non-terminals that are conditions of IF clauses?
> > > > * Why are multiple programs returned? I see Algorithm 1 returning only a single leaf node.
> > > > * What is the cross-validation score? The same as eq. (5)?
> > > > * Is limited depth encouraged by the structural cost ?
> > > > * Algorithm 1, line 3: $f(u)$ is not defined until later in line 10.
> > >
> > > * In this paper, all leaf nodes are considered as goal nodes. Leaf nodes contain only non-terminals from a DSL. We've updated Section 3 to clearly state that leaf nodes are goal nodes.
> > > * In this paper, the inputs of a substituted NN is always the feature vector $\mathbf{v}$ and the output is always a scalar. It is possible to design other kinds of domain-specific languages for which the input and output dimensionality of the substituted NN will be different in different contexts. For example, if the primitive $\alpha_1+\alpha_2$ returns a vector as output, the NNs substituted for $\alpha_1, \alpha_2$ must also return vectors as outputs. As said earlier, the NN that can be substituted for $\alpha_1$ in $\mathtt{if \ \alpha_1\ then\ \alpha_2\ else \alpha_3}$ takes the feature vector $\mathbf{v}$ as input and produces a scalar value as output. The resultant scalar value is then used to perform a conditional check using the smooth approximation of $\mathtt{if-then-else}$ as explained in Section 4.1. We've updated Sec 4.3 to make this point explicit.
> > > * We apologize for the confusion caused here. Algorithm 1 returns only one program $(\mathcal{P}^*, \theta^*)$ such that $(\mathcal{P}^*, \theta^*) = argmin_{(\mathcal{P}, \theta)} (s(\mathcal{P}) + \zeta(\mathcal{P}, \theta))$. We've updated algorithm 1 to reflect these changes. We've also updated the paragraph of Section 4.3 to explain how algorithm 1 uses cross-validation score to return the best program.
> > > * The cross-validation score does not depend on the structural cost. Cross-validation is performed similarly to traditional model training. That is, the model's parameters are chosen based on the cross-validation score. We've modified the paragraph of Section 4.3 to reflect these changes.
> > > * Yes. Since structural cost $s(\mathcal{P})$ of a program $\mathcal{P}$ is a part of the objective function in Equation 5, limited depth is encouraged by structural cost. Also, in our implementation, depth of the program can also be set as a hyperparameter to synthesize best program whose depth is with in a specified depth hyperparameter.
> > > * We defined $f(u_0):=\infty$ in line 1. When the line 3 gets executed for the first time, $Q$ contains only $u_0$ and hence $u$ gets the value $u_0$. Later, as in line 10 (please see revised version), a new $u$ will be added to $Q$. We hope this clarifies this doubt.
> > >
> > > >Section 5: Vagueness of theoretical statements
> > > > * Lemma 5.1: This lemma partly justifies the use of training loss as a valid heuristic function (underestimator of the true cost). But without specifying the value ....
> > >
> > > * We agree that the training loss value has an impact on approximating the heuristic value. In this paper we consider the notion of *$\epsilon$-admissibility* (as opposed to usual *admissibility*) that overcomes the following issues caused by the neural relaxations (Page 5 first paragraph of [Shah et al. NeurIPS 2020]).
> > >     * Neural networks used for substituting non-terminals in a partial program may only form an approximate relaxation of the space of completions and parameters of architectures.
> > >     * Training of these neural networks may not reach global optima due to various issues during training such as sub-optimal hyperparameters, number of epochs, etc.
> > >
> > > Even under such problems, the training loss can still be viewed as an $\epsilon$-admissible heuristic value. Following [Shah et al. NeurIPS 2020], we've slightly modified the statement of Lemma 5.1 in the revised version of the manuscript as shown below (bold text indicates the changes made).
> > >
> > > Lemma 5.1 (Neural Admissible Relaxations) In an informed search algorithm $\mathcal{A}$, given an internal node $u_i$ and a leaf node $u_l$, let the cost of the leaf edge $(u_i, u_l)$ be $ s( r)+\zeta(\mathcal{P}, \theta^*) $, where $\theta^{*} = argmin_{\theta} \zeta(\mathcal{P}, \theta)$ and $s(r)$ is the structural cost in using rule $r$ to create $u_l$ from $u_i$. If a neural network model $\mathcal{N}$ **with adequate capacity in terms of depth and width of $\mathcal{N}$** is used to substitute each non-terminal of $u_i$, the training loss of the program obtained is an $\epsilon-$admissible heuristic for $u_i$.
> > >
> > > Related references:
> > > `Shah, Ameesh, Eric Zhan, Jennifer Sun, Abhinav Verma, Yisong Yue, and Swarat Chaudhuri. "Learning differentiable programs with admissible neural heuristics." Advances in neural information processing systems 33 (2020): 4940-4952.`

---

> > > > ### Author Response · Authors · 2023-05-24
> > > > **Response to reviewer T5Zs (continued)**
> > > >
> > > > >Section 5: Vagueness of theoretical statements
> > > > >* Proposition 5.1: I find the statement of this result too vague.... statement beyond "there exists a DSL."
> > > >
> > > > We thank you for the suggestions and we agree with you. We've edited Proposition 5.1 in the revised version of the manuscript as follows.
> > > >
> > > > ***Proposition 5.1:** If the interventional effect of the treatment variable $T$ on the target variable $Y$ is a continuous function $g(T)$ i.e., $\mathbb{E}[Y|do(T)] =g(T)$, using any DSL $\mathcal{L}$ for synthesizing a single-hidden-layer neural network, NESTER synthesizes a program $(\mathcal{P},\theta)$ that $\epsilon-$approximates $g$ for a given $\epsilon > 0$.*
> > > >
> > > >
> > > > >The PEHE metric is not included in Table 3, and Appendix C only shows PEHE results for the IHDP dataset. I think it is important for completeness to report PEHE on the other datasets too, and ideally in a table in the main paper rather than the appendix since individual treatment effect (ITE) estimation is an important objective.
> > > >
> > > > We have added the Twins dataset results on $\epsilon_{PEHE}$ metric to the main paper in the revised version of the manuscript. As explained in Section 6, since we observe only one potential outcome in the Jobs dataset, we use $\epsilon_{ATT}$ metric for the results on Jobs dataset and hence $\epsilon_{PEHE}$ metric is not a valid measure for Jobs dataset.
> > > >
> > > > > I think there should be more discussion of how the synthesized programs, in Table 4 and in general, relate to specific existing NN architectures. Has NESTER come up with anything significantly different in the authors' experience? One particular aspect of this is how the complexity of NPS-generated estimators compares to those in the literature.
> > > >
> > > >
> > > > As explained in our response to the first question, the program generated for IHDP dataset (Table 4):
> > > >
> > > > $\mathtt{if\hspace{4pt} subset(\mathbf{v}, \{0..|\mathbf{v}|\})}$
> > > > $\mathtt{then\ transform(\mathbf{v})}$
> > > > $\mathtt{else\ transform(\mathbf{v})}$
> > > >
> > > > can be viewed as a two-headed neural network architecture such as TARNET,
> > > > CFR. The $\mathtt{transform(v)}$ primitive in $\mathtt{then, else}$ clauses can be viewed as the IPM regularization term implemented in CFR. Hence, the overall architecture shown above can be viewed as a CFR model.
> > > >
> > > > Similarly, the program structure generated for Jobs dataset (Table 4):
> > > >
> > > > $\mathtt{if\hspace{4pt} subset(\mathbf{v},\{0..|\mathbf{v}|\})}$
> > > > $\mathtt{then \hspace{4pt}\ subset(\mathbf{v},\{0..|\mathbf{v}|\})}$
> > > > $\mathtt{else\ \ subset(\mathbf{v},\{0..|\mathbf{v}|\})}$
> > > >
> > > > can be viewed as a simple two head neural network architecture without any additional regularization components because $\mathtt{subset(\mathbf{v},\{0..|\mathbf{v}|\})}$ simply acts as a multilayer perceptron that takes input features $\mathtt{v}$ as input and returns a scalar value as the potential outcome.
> > > >
> > > > NESTER outputs a simple yet different architecture compared to the existing two-head network architecture: $\mathtt{subset(\mathbf{v},\{0..|\mathbf{v}|\}))}$ which is a multi-layer perceptron that takes the input vector $\mathtt{v}$ as input and returns a scalar value as the potential outcome. In this case, the complexity of generated program by NESTER is much simpler than the existing models such as CFR, Dragonnet. There are few instances where we get the programs of the sort: $\mathtt{subset(\mathbf{v},\{0..|\mathbf{v}|\})\times subset(\mathbf{v},\{0..|\mathbf{v}|\})}$ for which there is no equivalent architecture in literature. The result of $\mathtt{subset(\mathbf{v},\{0..|\mathbf{v}|\})\times subset(\mathbf{v},\{0..|\mathbf{v}|\})}$ is simply the product of two multilayer perceptron's outcomes. Another interesting program generated by NESTER for IHDP (whose outputs are not best estimates of potential outcomes) is:
> > > >
> > > > $\mathtt{if\hspace{4pt} transform(\mathbf{v})}$
> > > > $\mathtt{then \hspace{4pt}\ transform(\mathbf{v})}$
> > > > $\mathtt{else\ \ transform(\mathbf{v})}$
> > > >
> > > > We believe that $\mathtt{transform}$ is not an ideal choice to be in conditional check clause and hence the reason for not giving optimal solutions.
> > > >
> > > > Overall, NESTER has the flexibility to generate both simple and complex architectures to solve a problem at hand.

---

> > > > > ### Author Response · Authors · 2023-05-24
> > > > > **Response to reviewer T5Zs (continued)**
> > > > >
> > > > > > Section 7, Choice of DSL: The ablation study seems incomplete.
> > > > > > * Why was primitive 4 (arithmetic operation) not considered for ablation?
> > > > > > * None of the returned programs in Table 4 used primitive 4, so why is combination 1,2,3 worse than 1,2,3,4?
> > > > > > * Why is combination 2,3,4 missing?
> > > > >
> > > > > We run each experiment 10 times and we show most frequently synthesized programs among those 10 runs. We infact use primitive 4 in our studies. The program below shows a program that uses multiplication as the arithmetic operation and constitutes to the average best score shown in Table 6 last row.
> > > > >
> > > > > $\mathtt{subset(\mathbf{v},\{0..|\mathbf{v}|\})\times subset(\mathbf{v},\{0..|\mathbf{v}|\})}$
> > > > >
> > > > > The purpose of this ablation study is to show that the primitives motivated from causal infrence literature achieve the best average scores with respect to the metrics considered.
> > > > >
> > > > > We apologize that we missed the entry for combination 2,3,4. We've added this entry to Table 6 of the revised version of the manuscript.

---

> > > > > > ### Author Response · Authors · 2023-05-24
> > > > > > **Response to reviewer T5Zs (continued)**
> > > > > >
> > > > > > **Minor comments and questions**
> > > > > > >Section 2, paragraph 1, paragraph 2 ...
> > > > > > * Yes, we agree. **'observed'** is a better word in this context compared to **'potential'**. We've changed it in the revised version of the manuscript.
> > > > > > * Yes, we agree. the phrase **'is the estimate of treatment effect'** improves readability in this context compared to **'is estimated as treatment effect'**. We've changed it in the revised version of the manuscript.
> > > > > > * Thank you for pointing it out. We've fixed this formatting issue in the revised version of the manuscript.
> > > > > > > Eq. (2): I believe the square should be inside...
> > > > > >
> > > > > > Thank you for pointing it out. We've moved the square inside the expectation of Equation 2. Changes are reflected in the revised version of the manuscript.
> > > > > > > Eq. (3): The notation $Y^0,Y^1$,  has already been defined...
> > > > > >
> > > > > > We use $Y^0, Y^1$ to majorly explain the concepts related to individual treatment effects. We use Pearl's $do(.)$ notation to explain concepts around average treatment effects. We believe the $do(.)$ notation is more familiar to many readers and will be helpful to better understand the paper. If the reviewer still thinks that having a single notation benefits the reader, we will make the changes to the manuscript accordingly.
> > > > > > >Section 4.1, "each of these listed primitives outputs a real scalar number"...
> > > > > >
> > > > > > It is possible to design one or all of the primitives to output a vector instead of a real number. As part of our design choice, we make all the primitives to output scalar values just because the overall output of a synthesized program is a scalar value. However, it is possible to design a new domain specific language (DSL) with some primitives producing scalar values and some primitives producing vectors. However, it is important to note that the final synthesized program should output a scalar value as we are predicting the potential outcome, a scalar value.
> > > > > > > Table 1, primitives 2 and 3
> > > > > > > * I wonder why "feed the resultant vector into an MLP" is part of ...
> > > > > > > * I think $\mathtt{transform}$ is too generic a term ...
> > > > > > * As explained in the previous response, it is our design choice. It is possible to design a new primitive that takes a vector as input and produces a scalar value. In such cases, we can make the $\mathtt{subset}$ and $\mathtt{tranform}$ primitives output vectors as output.
> > > > > > * Thank you for the suggestion. We will replace $\mathtt{transform}$ with $\mathtt{align}$ after the discussion period. Changing right away might confuse other reviewers.
> > > > > >
> > > > > > > Proposition 5.2: I think the real statement is that CFR..
> > > > > >
> > > > > > Yes, CFR is an instance of the programs that be synthesized using our method. For completeness, in proposition 5.2, we explicitly showed that error bounds of CFR naturally follow in our method.
> > > > > >
> > > > > > > Section 7, Interpretability of NESTER..
> > > > > >
> > > > > > We agree with you. We acknowledged this point in Section 7 as **'While this is not a surprising conclusion, the choice of such a $\mathtt{subset}$ primitive in the DSL allows us such an analysis'**. There can be cases where removing a some features (e.g., parents of treatment variables that do not belong to a backdoor path in the underlying causal graph [Cinelli et al. 2020]) can improve the precision in the estimation of treatment effect in finite sample analysis. Hence our analysis will be useful in analyzing such use cases. We've explained this point in the revised version of the manuscript.
> > > > > >
> > > > > > Related references
> > > > > > ```
> > > > > > Cinelli, Carlos, Andrew Forney, and Judea Pearl. "A crash course in good and bad controls."
> > > > > > ```
> > > > > > > Section 7, Runtime Analysis..
> > > > > >
> > > > > > Yes, the bulk of the complexity of NESTER accounts for training multiple architectures that are encountered during program tree expansion. For this reason, we limit the program depth to utmost 5 in main results. We observed that a depth of 5 was enough to get interpretable programs that can beat baselines.
> > > > > >
> > > > > > > Section 7, Analysis of Depth..
> > > > > > > * What does "take the first instance" mean?
> > > > > > > * Can the "better trade-off between in-sample and out-sample..
> > > > > > > * In the description of Figure 4 left panel....
> > > > > >
> > > > > > * The IHDP dataset contains 1000 realizations/datasets of simulated outcomes. Among those 1000 realizations/datasets, we take the first realization/dataset to verify the effect of program depth on $\epsilon_{ATE}$. We've replaced the word **instance** with **realization** in the revised version of the manuscript.
> > > > > > * Yes, by better trade-off, we mean that the sum of in-sample and out-sample is small when the depth of the program is four compared to other depths. We've added this point to the revised version of the manuscript.
> > > > > > * Thank you for bringing up this point. Yes, there are actually two depths (4,8) for which the sum of in-sample and out-sample $\epsilon_{ATE}$ values are much smaller than other depths. Choosing a program with depth 4 has the advantage of being simple, interpretable while achieving good results.
> > > > > >
> > > > > > We hope we've addressed all of your concerns. We would be happy to answer any further questions.

---

### Review · Reviewer_z8ko · 2023-05-08

**Summary Of Contributions:**

The paper proposes NESTER, a method of creating architectures for estimating the (causal) average treatment effect by introduction of a domain-specific language (DSL) that neurosymbolic program synthesis can be applied to. This DSL can express several prior approaches but is not limited to that.

The paper examines the theoretical properties of their approach, presenting a universal approximation result, and compares the empirical performance to various baselines.

**Audience:**

Yes

**Broader Impact Concerns:**

Given the topic, adding a broader impact statement (or a short paragraph in the conclusions) might be good. To at least shortly mention any potential ethical concerns or biases associated with using NESTER in real-world scenarios, which could be caused by biased observational data or situations where the assumption do not hold. (But again, I'm not an expert in this area, so maybe, that's all well-known?)

**Claims And Evidence:**

Yes

**Requested Changes:**

Answering Q 1, 2, and 3 is important for my acceptance.

Q 4 and 5 would be great to have addressed in the paper as other readers might have similar questions.

**Strengths And Weaknesses:**

As a disclaimer, I'm not very well-versed in the literature on causal inference. I hope the authors and other reviewers will excuse and point out any mistakes.

The paper is very well-written. I particularly like the exposition and introduction of various properties and common assumptions in §3, which has provided an informative introduction.

§4 and §5 provide a good treatment of NESTER. The DSL is introduced using its production rules, which are then tied to concepts from prior art and neural network architectures.

The paper compares NESTER to 16(!) baselines with great results and provides several ablations to examine the properties of their method.

That said, I have several questions which are connected to my requested changes below:

1. In Figure 1, for DragonNet, how does the DSL express $p(t \mid \phi)$? I am not acquainted with DragonNet, but assuming this head of the model is also trained, it might provide benefits---so even though NESTER outperforms it in the experiments---is it correct to say that NESTER can generate DragonNets? (ie, it would not seem that an equivalence is possible?)

2. How does NESTER relate to neural architecture search approaches? NPS seems to be related to it. Could this be addressed in the related work?

3. What are the limitations of NESTER? It would be good to discuss these in the conclusion.

4. In Table 3, NESTER performs better for Out-of-Sample than In-Sample on the Jobs dataset. How is that possible or reasonable? I see that other baselines have similar results, but I find it curious that a method can perform better out-of-sample.

5. For Prop 5.2.: does this essentially say that because NESTER can express the architecture of CFR, it thus is at least as good as it?

Overall, the contributions are evidenced, and people in the field will find the paper interesting.

---

> ### Author Response · Authors · 2023-05-24
> **Response to reviewer z8ko**
>
> Thank you for your insightful review. We address each of your concerns below. We've also uploaded the revised version of the manuscript addressing the concerns.
>
> > In Figure 1, for DragonNet, how does the DSL express $p(t|\phi)$?...is it correct to say that NESTER can generate DragonNets?
>
> As explained in Sec 4.2, in Dragonnet, the head $p(t|\phi)$ is trained to achieve propensity score matching, i.e. to control for parents of the treatment variable. That is, training the head $p(t|\phi)$ forces the learned representation $\phi$ to contain the information about the parents $\mathtt{pa_T}$ of $T$. Since we do not assume knowledge of the underlying causal graph, we rely on $\mathtt{subset(\mathbf{v},S)}$ primitive with various sets $\mathtt{S}$ to identify parents of $T$. That is, for a particular (unknown) set $\mathtt{S}$ of indices, we get the features of $\mathtt{pa_T}$ from $\mathbf{v}$. In our framework, the program synthesizer comes up with an appropriate set $S$ that gives the best performance (please see Figure 3). Using this idea, the two heads of the program $\mathcal{P}_D$ shown in Figure 1 use $\mathtt{pa_T}$ to estimate potential outcomes, similar to Dragonnet. We note that architecturally Dragonnet and $\mathcal{P}_D$ can be different but functionally they are similar. This has been highlighted in the caption of Figure 1.
>
> > How does NESTER relate to neural architecture search approaches? NPS seems to be related to it. Could this be addressed in the related work?
>
> The significant difference between neural architecture search (NAS) and NPS is that, in NPS, symbolic and domain knowledge can be introduced in terms of program primitives of the domain-specific language. However, the goal in NAS is to design the best-performing architecture using a combination of standard neural network components, such as convolution blocks. In NPS, the domain-specific language can change for different applications/domains, based on available domain knowledge; whereas in NAS, the underlying neural network components are generally fixed. NPS can be viewed as combining symbolic reasoning (which NAS methods don't focus on) and representation learning algorithms, making it a good choice for treatment effect estimation.
>
> We have added this discussion as a new paragraph in the related work section of the revised manuscript.
>
> > What are the limitations of NESTER? It would be good to discuss these in the conclusion.
>
> The number of primitives in current DSL (Table 1) are only four. The number of primitives can be increased to encode additional inductive biases from causal inference literature. Another limitation is the time required to synthesize programs. However, we observed that the programs with small depths are enough to synthesize programs similar to existing methods and achieve state-of-the-art performance. As per the suggestion from reviewer MQtc, we performed additional experiments using a different program synthesis method called dPads [1] to address time complexity issues. We've already mentioned the first point about the extensibility of DSL in the conclusions and we've added the second point on time complexity to the conclusions in the revised manuscript.
>
> Related references
> [1] https://github.com/RU-Automated-Reasoning-Group/dPads
>
> > In Table 3, NESTER performs better for Out-of-Sample than In-Sample on the Jobs dataset. How is that possible or reasonable? I see that other baselines have similar results, but I find it curious that a method can perform better out-of-sample.
>
>
> In traditional supervised learning algorithms, we usually observe that in-sample performance is better than out-sample performance. However, as stated in Section 6 --> Evaluation Metrics, since we do not observe all potential outcomes during training, in-sample performance becomes non-trivial in treatment effect estimation, and thus, in-sample results may be worse than out-sample results because of unseen potential outcomes during training. We observe this pattern not only in Jobs dataset but also in IHDP, Twins datasets for other baseline methods as well (see BNN results in Table 9 of the Appendix in the original submission and Table 4 of the main paper in the revised version). In Table 3 of the revised version, we see that results w.r.t. $\sqrt{\epsilon_{PEHE}}$ has a similar pattern (BNN on IHDP, Twins; OLS-1,TARNet,CFR, C Forest, BART on Twins)
>
> >For Prop 5.2.: does this essentially say that because NESTER can express the architecture of CFR, it thus is at least as good as it?
>
> Yes, NESTER is a generalization of existing multi-head neural network architectures for treatment effect estimation. Hence, CFR, a two-head neural network architecture with IPM regularization, is a special case of NESTER. Since NESTER searches for an optimal program from a set of synthesized programs, NESTER is functionally at least as good as CFR.
>
> We hope we've addressed all of your concerns. We would be happy to answer any further questions.

---

### Review · Reviewer_MQtc · 2023-05-09

**Summary Of Contributions:**

Previous research on estimating treatment effects from observational data has focused on developing deep neural network models that are specifically designed for certain causal discovery problems. As a result, different neural network models were suitable for different causal graphs. If the causal graph was not known a-priori, practitioners had to try out various architectures to find the most appropriate one. To overcome this challenge, the authors propose treating treatment effect estimation as a search problem by using a domain-specific programming language (DSL) to represent a distribution of architectures. They leverage an off-the-shelf program synthesizer to identify the best-performing architecture based on the dataset and DSL. The authors evaluate their approach against several baselines using three standard datasets and demonstrate that their tool, called NESTER, achieves slightly better performance than the baselines on two datasets and matches the best baseline's performance on the third dataset.

**Audience:**

Yes

**Broader Impact Concerns:**

The manuscript doesn't seem to contain a broader impact statement section. I do not believe the methods presented in this paper raise serious moral or ethical concerns.

**Claims And Evidence:**

Yes

**Requested Changes:**

Both of my concerns inherently stem from the lack of evaluation on domains which require complex programs in the DSL. Would it be possible to synthetically generate a treatment effect estimation dataset which requires a program of depth 2 or more to solve it? Affirmative results on this would address both my concerns.

Alternatively, I have suggested some algorithmic changes that may yield further improvements:
- _(Regarding Weakness 1)_ Related work in using DNN for ITE estimation [2] seems to have shifted to neural networks to avoid the NP hard cost of learning causal graphs directly. However, the current formulation of program synthesis reintroduces the combinatorial search problem. To address this, I propose that the authors investigate recent work in amortized search for program synthesis[3][4]. Under the amortized search framework, we train a neural network to learn to search for programs at train time. This avoids the cost of searching over a space of possible programs. However, this approach usually requires more data than NEAR and a diverse set of tasks.
- _(Regarding Weakness 2)_ I recommend the authors to look into dPads[1]. dPads formulates program synthesis as an architecture search problem and proposes a fully differentiable algorithm for neurosymbolic program synthesis. It avoids a lot of the pitfalls the authors encountered with NEAR (pathological behavior. Point [1] I made earlier). The dPads codebase follows the same format as the NEAR codebase, so it shouldn't be hard to shift to dPads. This might further improve the results compared to baselines.


[1] https://github.com/RU-Automated-Reasoning-Group/dPads

[2] https://arxiv.org/abs/1906.02120

[3] https://arxiv.org/abs/2203.10452

[4] https://arxiv.org/abs/2206.05922

**Strengths And Weaknesses:**

Strengths:
* To the best of my knowledge, this is the first paper that proposes using program synthesis for treatment effect estimation (using DNNs). This is a novel contribution and useful if the causal graph is small and not known a priori.
* Neurosymbolic program synthesis allows for a degree of interpretability of the generated models.
* The presentation is clear and easy to follow along.
* The supplementary material is well organized, and I was able to run the code without any runtime errors (I did not attempt to replicate results).

Weaknesses:
1. Program synthesis involves a combinatorial search over a distribution of programs. I'm not convinced that the slightly better results are worth the complexity tradeoff of finding a program within the DSL; especially when the ground truth program is large. This, in my view, severely limits the impact of this work for the causal discovery audience.
2. The DSL seems to admit trainable MLPs for the `transform` primitives. From personal experience, NEAR does not work well when the domain specific language admits neural networks in intermediate nodes and the domain necessitates nested compositions for representing complex programs [1]. I'm concerned that this approach will not generalize to large, complex ground truth programs / causal graphs.

[1] Consider the following: assume we need to model a large causal graph which necessitates repeated calls to parametrized DSL functions (ie: $f_{\theta_1}(f_{\theta_2}(\dots(f_{\theta_n}(input))))$. If each function has an error $\epsilon_i$ associated with calling it, intuitively, the error bound of the programs become multiplicative larger with each nested function call. NEAR is a top-down program synthesizer, which means it has to rate partial programs by filling the "hole" with a type-appropriate neural network and calculating the loss. However, at a depth of $i \in (0, n)$, it's possible that the error bounds become large enough that all potential paths are equally bad.  Another way of looking at this: NEAR's $\epsilon$-admissibility guarantee only takes into account the parametric function used to rate the partial programs, and is DSL agnostic. If the DSL contains parametric functions on only the terminal nodes, NEAR still performs relatively well. However, if the DSL contains parametric functions in the non-terminal nodes, the error propagates and NEAR's performance starts decreasing.

---

> ### Author Response · Authors · 2023-05-24
> **Response to reviewer MQtc**
>
> Thank you for your insightful and encouraging review. We address each of your concerns below. We've also uploaded the revised version of the manuscript addressing the concerns.
>
> We would like to begin by highlighting that our proposed method does not depend on the size of the underlying causal graph. Also, we are different from causal discovery literature and we do not perform causal discovery in this work. As discussed in the Related Work section-->Relevance of Causal Discovery Methods, we avoid performing causal discovery (similar to existing methods that estimate treatment effect using learning-based methods). Our goal is to estimate treatment effects given observational data alone by generalizing existing multi-head neural network-based methods. Given this context, we now address your concerns below.
>
> > Program synthesis involves a combinatorial search over a distribution of programs. I'm not convinced that... severely limits the impact of this work for the causal discovery audience.
>
> As discussed earlier, our method is not intended for a causal discovery audience. We believe your concern was about handling a combinatorial search over a huge set of programs and the possibility of getting large synthesized programs. We agree with you on this, and do hope that such a work will also promote future efforts for scalability and efficiency. However, from our own studies, we observe that small program depths lead to simple programs that are functionally similar to existing neural network based methods on treatment effect estimation and are interpretable. For this reason, we limit the program depth to utmost 5 in our main results. We observed that a depth of 5 is enough to get interpretable programs that can beat baselines in terms of performance. We thank you for sharing the related references [3],[4] on the amortized search for program synthesis. We've cited these papers in the revised version as they are relevant to neurosymbolic program synthesis. We also thank you for sharing dPads [1]. We discuss about dPads in our next response.
>
> > The DSL seems to admit trainable MLPs for the $\mathtt{transform}$ primitives. From personal experience, NEAR does not work well.... will not generalize to large, complex ground truth programs / causal graphs.
>
> Thank you for the detailed explanation of problems with neural admissible relaxations (NEAR) paper with an example and sharing the information about dPads [1]. We again emphasize that our method doesn't depend on the complexity of ground truth causal graphs. However, we followed your suggestion and implemented our method using dPads [1], that avoids combinatorial search using differentiable synthesis. We observed improved results compared to NEAR method in some settings as shown in the tables below. We observe a lot of improvement at runtime as shown in the results below. We've added these results to the revised version of the paper. These new results strengthen our paper in terms of performance comparison over baselines. Code for NESTER-dPads is provided in the updated supplementary materials.
>
>
> | Method       | IHDP (In-Sample $\epsilon_{ATE}$) | IHDP (Out-Sample $\epsilon_{ATE}$) | Twins (In-Sample $\epsilon_{ATE}$) | Twins (Out-Sample $\epsilon_{ATE}$) | Jobs  (In-Sample $\epsilon_{ATT}$) | Jobs (Out-Sample $\epsilon_{ATT}$) |
> | ------------ | ---------------- | ----------------- | ----------------- | ------------------ | ----------------- | ----------------- |
> | NESTER-NEAR  |    **0.05 $\pm$ 0.04**              |     **0.05 $\pm$ 0.03**              |         **0.0034 $\pm$ 0.0005**          |       0.0039 $\pm$ 0.0006             |          0.06 $\pm$ 0.00          |      0.02 $\pm$ 0.01              |
> | NESTER-dPads |     **0.05 $\pm$ 0.01**             |    **0.05 $\pm$ 0.02**               |      0.0043 $\pm$ 0.0001            |    **0.0028 $\pm$ 0.0001**                |        0.06 $\pm$ 0.00           |    **0.01 $\pm$ 0.01**              |
>
>
>
> | Method       | IHDP (In-Sample $\sqrt{\epsilon_{PEHE}}$) | IHDP (Out-Sample $\sqrt{\epsilon_{PEHE}}$) | Twins (In-Sample $\sqrt{\epsilon_{PEHE}}$) | Twins (Out-Sample $\sqrt{\epsilon_{PEHE}}$) |
> | ------------ | ---------------- | ----------------- | ----------------- | ------------------ |
> | NESTER-NEAR  |    0.73 $\pm$ 0.19              |    **0.76 $\pm$ 0.20**               |         0.318 $\pm$ 0.002          |   **0.319 $\pm$ 0.000**                 |
> | NESTER-dPads |    **0.71 $\pm$ 0.10**              |    **0.76 $\pm$ 0.32**               |     **0.314 $\pm$ 0.001**   |     0.331 $\pm$ 0.002  |
>
>
> Run time (in minutes) analysis
>
> | Dataset | SNet | NESTER-NEAR | NESTER-dPads |
> | ------- | ---- | ----------- | ------------ |
> | Twins | 1.85 $\pm$ 0.3|   2.12 $\pm$ 0.12   |  **1.40 $\pm$ 0.20**|
> | Jobs   | 1.23 $\pm$ 0.2 |   1.09 $\pm$ 0.40| **1.00 $\pm$ 0.10**|
>
> We hope we've addressed all of your concerns. We would be happy to answer any further questions.

---

> > ### Comment · Reviewer_MQtc · 2023-06-04
> > **Followup**
> >
> > **Rebuttal Follow-up**
> >
> > Thank you for the clarification on the goals of this paper.  Thank you for running the additional experiments as well. I’m glad that dPads is faster on this problem domain as well.
> >
> > * I still hold some concerns about the scalability of NESTER. However, I agree with the authors that NESTER’s DSL design allows it to avoid the scalability problems in practice. Specifically, NESTER’s DSL admits abstractions that allow viable architectures to be represented as small programs And these small programs can be found without exploring a large search space. However, as additional heuristics are baked into the DSL, the program search will re-emerge as a bottleneck. Yet, this problem can be relegated to future work.
> > * Since the search space is small, simply enumerating all possible programs is a feasible search strategy here. For the sake of completeness, I recommend the authors also attempt naive enumeration for this problem domain (with a reasonable timeout, say 5 minutes) instead of A-star.  NEAR’s codebase already implement the enumeration algorithm.
> > * I’d be curious to know why there is a discrepancy between NEAR and dPads’ performance on the in-distribution Twins dataset.
> >
> > Overall, I recommend borderline acceptance. The authors’ attempt to bridge treatment affect estimation and neurosymbolic program synthesis by proposing a DSL that allows off-the-shelf program synthesizers to be used for treatment effect estimation. Their approach leaves much to be desired: the DSL is fairly primitive, and the program synthesis is not tightly integrated into the DSL design. I cannot comment on the correctness of the DSL, as I’m not a domain expert. Yet, in evaluations, NESTER is able to achieve on-par performance, which makes me hopeful that future work in improving the DSL and the program synthesis will enable better performance.

---

### Review · Reviewer_W2DP · 2023-05-09

**Summary Of Contributions:**

- The paper proposes the use of neurosymbolic program synthesis (NPS) for treatment effect estimation. This is achieved by defining a domain-specific language (DSL) of common architecture design patterns from the literature and building program trees using these primitives. The approach is evaluated on known benchmarks for treatment effect estimation and compared to a large number of baselines.

- The claimed contributions are as follows:
  * Development of an adaptive neurosymbolic method for estimating treatment effects from observational data
  * A domain-specific language
  * A study of the universal approximation ability of the neurosymbolic programs
  * Comprehensive empirical studies on multiple benchmark datasets

**Audience:**

Yes

**Claims And Evidence:**

No

**Requested Changes:**

- A revision should address the weaknesses listed above.

**Strengths And Weaknesses:**

# Strengths

- The use of neuropsymbolic program synthesis for treatment effect estimation is interesting and is, to my knowledge, novel. This could be of interest to the TMLR audience.
- The empirical study is large in scope, including many established baseline methods

# Weaknesses

- The description of the programs and the DSL are very informal and hard to follow. I don't think the results could be reproduced based on the current version of the paper. For example, the clause "if subset(v, {0})" is ambiguous since 4.1 describes the subset primitive as returning a real number. What is the boolean value of this number? The same is true for the other primitives, such as transform(v), which has many unknown parameters including the discrepancy used for minimising cohort differences. Is the use of MMD hard-coded? Are the groups always split by the variable "t"?

- The paper relies very heavily on existing theoretical results to justify its approach (section 5), which amounts to arguing that E[Y | X, t] can be modeled using NPS. Hence, the theoretical analysis is rather light and the main contribution of this work is to identify existing patterns in neural architectures for treatment effect estimation, translating them to a DSL, and evaluating this heuristic empirically.

- The authors argue that a benefit of their setting is that they do not need to perform causal discovery to identify a causal graph and then the treatment effect. Instead, they rely on the ignorability assumption. First, this is true for all of the baselines used, who also assume that ignorability is satisfied. Second, it is not clear how the authors _justify_ the ignorability condition without the causal graph. Commonly, in practice, the causal graph would not be derived by a causal discovery algorithm but based on domain knowledge. Ignorability follows as a consequence. Domain knowledge would have to be used to justify ignorability without the graph as well. In either case, this assumption is not at all specific to the method used here. Moreover, the authors remark that in the ignorability framework, "all observed features are to be controlled". This has two problems: 1) the observed features may not be sufficient and 2) controlling for some of them may introduce confounding bias (e.g., M-bias) rather than reduce it.

- In 4.1, it is stated that "transform(v)" acts like a balancing representation from Shalit et al and the statement is made that "Given two inputs v_0 and v_1, we would want \phi(v_0) \approx \phi(v_1)". This is not at all what the MMD seeks to accomplish since this representation would be essentially information less. The MMD penalizes distributional differences, not point-wise differences. Moreover, in Figure 1, CFR is compared to a program which transforms the inputs using different functions depending on treatment status. This is not at all what CFR does since the representation is shared for both treatment groups. Finally, if the "transform" primitive balances the treatment groups after the the subset() operation, wouldn't there only be samples from a single treatment arm in each transform? How would the MMD be computed in this case?

- The main paper only presents the ATE results from IHDP but the Appendix contains results that shows that the ITE/CATE performance is at best comparable with existing results.

- The evaluation is performed on well-known datasets. This has its strengths, but it is also the case that these datasets are rather small which makes it more difficult to automate the architecture search based on cross-validation.

## Minor weaknesses and questions

- Definition 3.1. is referred to as the "individual treatment effect". While this has been used by some notable works, it is generally recognised that "conditional average treatment effect" is a more appropriate term since individual treatment effect is a counterfactual quantity in the Pearlian sense. Hence, I believe the Pearl citation before (1) is incorrect.

- The ATE is defined using do-notation but ITE with potential outcomes. It is not clear why this choice is made.

- Assumptions 3.1 are stated without an identifiability result following. I suspect what the authors want is to argue that E[Y | T=t, X=x] = E[Y | do(T=t), X=x] as is customary.

- The notation pa_T used in Figure 1 for the Dragonnet comparison is never defined.

- The abstract refers to "all the desiderata". What are these?

---

> ### Author Response · Authors · 2023-05-24
> **Response to reviewer W2DP**
>
> Thank you for your insightful review. We address each of your concerns below. We've also uploaded the revised version of the manuscript that addresses the concerns.
>
> > The description of the programs and the DSL are very informal and hard to follow... Are the groups always split by the variable "t"?
>
> We apologize for the confusion caused. As explained in Section 4.1, we implement a smooth approximation of the primitive $\mathtt{if\ \alpha1\ then\ \alpha_2\ else\ \alpha3 }$. For this reason, it is not required for $\alpha_1$ ($\alpha_1$ can be any primitive) to return a boolean value. The real number returned by $\alpha_1$ is used in the smooth approximation expression. We've added a line to the paragraph after Table 1 in the revised manuscript to clarify this. As discussed in another response to your question about $\mathtt{transform}$, we pre-train the MMD module in $\mathtt{transform}$ primitive so that all instances of $\mathtt{transform}$ in a synthesized program use the same learned representation $\phi$, making it functionally similar to CFR. We've updated Table 1 and the paragraph describing $\mathtt{transform}$ primitive to highlight the **pre-training** aspect of $\phi$ in the $\mathtt{transform}$ primitive. If $\mathtt{subset(\mathbf{v}, S)}$ is used in the $\mathtt{if}$ clause, the group split is based on the set of indices $\mathtt{S}$. Since the set $\mathtt{S}$ can take on any set of indices, it is not always guaranteed that the split is based on $t$. The examples given in Table 1 are  programs that are functionally similar to existing methods, showcasing the generalizability of NESTER. The actual programs returned by our method are shown in Table 5 of the revised version (Table 4 in the earlier version).
>
> > The paper relies very heavily on existing theoretical results to justify its approach (section 5)... and evaluating this heuristic empirically.
>
> We agree with you. While our main contributions are on proposing a domain-specific language (that in turn allows us to use neurosymbolic programming approach for treatment effect estimation) and the empirical studies, we believe that explicitly showing the theoretical connections of our approach with existing results and thus showing the ability of NPS for treatment effect estimation helps guide further research on using NPS for treatment effect estimation.
>
> > The authors argue that a benefit of their setting is that they do not need to perform causal discovery to identify a causal graph and then the treatment effect. Instead, they rely on the ignorability assumption.... This has two problems: 1) the observed features may not be sufficient and 2) controlling for some of them may introduce confounding bias (e.g., M-bias) rather than reduce it.
>
> We do **not claim/argue** that not performing causal discovery is a benefit of our method. We rather state that, similar to other baselines in our paper, our method is based on the ignorability assumption. We assume ignorability/no-latent-confounding to ensure that the set of features to control is within the set of observed features (i.e., no unobserved confounding). Later, similar to existing methods, we rely on our primitives to identify the variables to control (e.g., Dragonnet uses the head $p(t|\phi)$ to learn about the parents of the treatment variable to estimate potential outcomes, which we simulate using the $\mathtt{subset}$ primitive.)
>
> In response to your final point, we acknowledge our previous error in asserting the necessity of controlling all observed features under ignorability. To rectify this incorrect statement, we have revised the third paragraph of the introduction to include the following line.
>
> **...under the no latent confounding/ignorability assumption, methods based on the classical Neyman-Rubin potential outcomes framework assume that a known set of observed features to control is available.**

---

> > ### Author Response · Authors · 2023-05-24
> > **Response to reviewer W2DP (continued)**
> >
> > > In 4.1, it is stated that $\mathtt{transform(v)}$ acts like a balancing representation from Shalit et al and the statement is made that "Given two inputs $v_0$ and $v_1$, we would want $\phi(v_0) \approx \phi(v_1)$". This is not at all what the MMD seeks to accomplish since this representation would be essentially information less.... How would the MMD be computed in this case?
> >
> > Apologies for the confusion caused here. The answer for all of your questions in this context is that the MMD module in $\mathtt{transform}$ is **pre-trained** to penalize distributional difference between the distributions $p(x|t=0)$ and $p(x|t=1)$. Recall from Table 1 that $\mathtt{transform}$ is a combination of a transformation through $\phi$ and an MLP. Now, the primitive $\mathtt{transform}$ appearing in both heads of the program $\mathcal{P}_C$ has the same underlying pre-trained model $\phi$, making it functionally similar to CFR. That is, since $\phi$ is pre-trained, the representation is shared between two heads of the program $\mathcal{P}_C$. Also note that the MLP part of the $\mathtt{transform}$ is not pre-trained and hence the final output of $\mathtt{transfrom}$ is different for different heads.
> >
> > We've updated these points in the revised version of the paper (Table 1 and the paragraph explaining $\mathtt{transform}$ in Section 4.1).
> >
> > > The main paper only presents the ATE results from IHDP but the Appendix contains results that show that the ITE/CATE performance is at best comparable with existing results.
> >
> > In the revised version of the paper, we've moved the ITE/CATE results from Appendix to the main paper. We've also added the results on the twins dataset ITE/CATE results in the revised version.
> >
> > > The evaluation is performed on well-known datasets. This has its strengths... difficult to automate the architecture search based on cross-validation.
> >
> > We followed existing work to experiment on  well-known datasets. Even if datasets are small, we observe that NESTER achieved better results in many cases. Our studies showed that small program depths lead to simpler programs that are also functionally similar to existing methods. We hence believe that small programs avoid the difficulty of automating the architecture search based on cross-validation (which is also in line with our own experimental setup).
> >
> > > Definition 3.1. is referred to as the "individual treatment effect"... Hence, I believe the Pearl citation before (1) is incorrect.
> >
> > Thank you for pointing this out. We've removed the Pearl citation before Definition 3.1 (Equation 1) in the revised version of the manuscript.
> >
> > > The ATE is defined using do-notation but ITE with potential outcomes. It is not clear why this choice is made.
> >
> > We use $Y^0, Y^1$ to majorly explain the concepts related to individual treatment effects. We use Pearl's $do(.)$ notation to explain concepts around average treatment effects. We believe the $do(.)$ notation is more familiar to many readers and will be helpful to better understand the paper. If the reviewer thinks that having a single notation benefits the reader, we will make the changes to the manuscript accordingly.
> >
> > > Assumptions 3.1 are stated without an identifiability result following. I suspect what the authors want is to argue that E[Y | T=t, X=x] = E[Y | do(T=t), X=x] as is customary.
> >
> > Yes, under assumptions 3.1, the treatment effect is identifiable. We think it would be trivial to state the identifiability result in this case. If it is useful, we will add an identifiability result in the revised version of the manuscript.
> >
> > > The notation $\mathtt{pa_T}$ used in Figure 1 for the Dragonnet comparison is never defined.
> >
> > Thank you for pointing it out. We've updated the caption of Figure 1 in the revised version to introduce $\mathtt{pa_T}$.
> >
> > > The abstract refers to "all the desiderata". What are these?
> >
> > By desiderata, we mean propensity matching, enforcing randomization etc. as mentioned in the abstract. To make it clear, we've updated the corresponding line in the abstract as follows in the revised version of the manuscript.
> >
> > **NESTER brings together the ideas used in existing methods based on multi-head neural networks for treatment effect estimation into one framework.**
> >
> > We hope we've addressed all of your concerns. We would be happy to answer any further questions.

---

### Author Response · Authors · 2023-05-24
**Common response to all reviewers**


We thank all reviewers for their thoughtful feedback. We are motivated to see that the reviewers found our paper is written well (MQtc, z8ko), our method is novel and useful to the community (MQtc, z8ko, W2DP), and experiments are thorough and complete (z8ko, T5Zs, W2DP). We've uploaded a revised paper incorporating all the reviewers' suggested changes. We've also uploaded a second revised version showing the changes made in red color for the convenience of reviewers. We've added a broader impact statement in the conclusions. We now address each reviewer's concerns below.

---

### Decision · Action_Editors · 2023-06-06

**Recommendation:** Reject

**Comment:**

During the discussion period there wasn't a clear consensus among the four reviewers, but ultimately none of them recommended a clear acceptance. Two of the reviewers recommended borderline acceptance, although they acknowledged a lower-confidence review. Another reviewer recommended borderline rejection, while the last one recommended rejection.

Taking all of this into account, unfortunately I cannot recommend acceptance of the paper in its current form. The numerous corrections and lack of clarity indicate that a second full round of reviews would be more appropriate. Because of that, I strongly encourage the authors to resubmit to TMLR.

On a more positive note, the reviewers praised the ideas in the paper (specifically, the novelty of using program synthesis for treatment effect estimation) and the empirical study (specifically, the numerous established baselines that the authors considered). Therefore I think that taking into account the reviewers' comments and improving the overall clarity of the manuscript would make it warrant acceptance at TMLR.

**Audience:**

The reviewers acknowledged that the method is novel and useful to the research community.

**Claims And Evidence:**

This paper addresses the problem of treatment effect estimation by using domain-specific programming language (DSL) to represent a distribution of neural network architectures. Given a dataset, the paper leverages an off-the-shelf program synthesizer to identify the best-performing architecture for that dataset. The method is evaluated on three standard benchmarks and against several baselines, showing slightly better performance.

The authors made an extensive set of updates after the reviews, even incorporating a new baseline (namely, dPads). Despite that, one of the main concerns raised by the reviewers is that the paper is still missing the required level of clarity to support the claims. There are a few points that raised confusion among the reviewers (e.g., the smooth approximation of the if-then-else and its associated temperature parameter, to name one).


**Resubmission Of Major Revision:**

The authors may consider submitting a major revision at a later time.